# Cycle Conditioning for Robust Representation Learning from Categorical Data

**Mohsen Tabejamaat**     *mohsen.tabejamaat@hh.se*
*School of Information Technology, Halmstad University, Sweden*

**Farzaneh Etminani**     *farzaneh.etminani@hh.se*
*Centrum for Research and Innovation, Region Halland, Sweden*
*School of Information Technology, Halmstad University, Sweden*

**Mattias Ohlsson**     *mattias.ohlsson@cec.lu.se*
*Centre for Environmental and Climate Science, Lund University, Sweden*
*School of Information Technology, Halmstad University, Sweden*

**Reviewed on OpenReview:** *https://openreview.net/forum?id=GkYOcbNLaW*

## Abstract

This paper introduces a novel diffusion-based method for learning representations from categorical data. Conditional diffusion models have demonstrated their potential to extract meaningful representations from input samples. However, they often struggle to yield versatile, multi-task information, limiting their adaptability to unforeseen tasks. To address this, we propose a cycle conditioning approach for diffusion models, designed to capture expressive information from conditioning samples. However, cycle conditioning alone can be insufficient. Diffusion models may ignore conditioning samples that vary across training iterations, an issue that occurs within cycle conditioning. To counter this limitation, we introduce additional "spelling" information to guide the conditioning process, ensuring that the conditioning sample remains influential during denoising. While this supervision enhances the generalizability of extracted representations, it is constrained by the sparse nature of spelling information in categorical data, leading to sparse latent conditions. This sparsity reduces the robustness of the extracted representations for downstream tasks or as effective guidance in the diffusion process. To overcome this challenge, we propose a linear navigation strategy within the latent space of conditioning samples, allowing dense representations to be extracted even with sparse supervision. Our experiments demonstrate that our method achieves at least a 1.42% improvement in AUROC and a 4.12% improvement in AUCPR over the best results from existing state-of-the-art methods.

## 1 Introduction

A categorical time series is a sequence of observations where each observation corresponds to a specific category rather than a numerical value, making interpolation between observations neither possible nor meaningful. Healthcare trajectories are a prominent example, often consisting of large categorical time series with observations representing medical activities, like prescribed medications or treatment procedures. Effectively utilizing such sequences requires creating compact and meaningful representations of these observations through a process called representation learning. This step is crucial for improving the accuracy and computational efficiency of downstream tasks. While unsupervised representation learning has proven effective for continuous data (Le-Khac et al., 2020; Poudel et al., 2024; Jang & Wang, 2023; Izacard et al., 2021; Caron et al., 2020), the need for various augmentations has made it difficult to benefit from this unsupervised strategy in learning representation for categorical sequences. For categorical data, augmentation

is mostly limited to *token masking*[1], which helps with missing tokens (part or whole of an observation) but fails to capture any complex relationships between the tokens. This limitation reduces the effectiveness of unsupervised representation learning for categorical sequences.

Utilizing representations learned within the latent space of generative models offers a promising alternative that eliminates the need for data augmentation in representation learning tasks (Afkanpour et al., 2024). In this approach, a generative model is first trained to accurately capture the underlying distribution of the data. The representation of input samples is then derived from their projection onto the bottleneck layer of the generative model (Abdal et al., 2019; Radford et al., 2015; Zhang et al., 2022a). Recently, diffusion models have demonstrated significant effectiveness in achieving high-quality fits to data distributions, which has led researchers to investigate their potential in representation learning (Yang et al., 2024; Hudson et al., 2024; Preechakul et al., 2022). However, as these models do not include a bottleneck layer, attention has shifted to conditional diffusion models (Yue et al., 2024). These models allow for projecting a conditioning sample into a latent space, which can then be used to guide the denoising process of the diffusion model. While conditioning effectively directs the generative process toward specific classes of samples, it does not inherently guarantee the extraction of versatile information from the conditioning samples. Consequently, the model may overlook essential, multi-task information within the conditioning samples that is necessary for good performance of these representations in an unforeseen downstream task.

In addition, these methods have primarily been developed for continuous visual data, which aligns with the continuous nature of diffusion models . The forward process of diffusion models operates on the assumption that adding Gaussian noise to a noisy sample produces a result that can be sampled from a Gaussian distribution. This implies that operations in the latent space of diffusion models occur within a continuous framework. As a result, these methods cannot be directly applied to representation learning from categorical datasets. Although there have been attempts to adapt diffusion processes for discrete data distributions (Hoogeboom et al., 2021; Austin et al., 2021), extending these methods to incorporate a conditioning sample remains challenging, particularly when the condition is derived from the output layer of an encoder, which is inherently continuous. This difficulty limits the applicability of these approaches in tasks such as representation learning.

In this paper, we show that cycle conditioning in diffusion models can effectively extract multi-task representations from categorical time series. To address the mismatch between categorical data and continuous diffusion models, we propose a learnable linear transformation that converts each categorical series into a continuous representation. This transformation does not require feedback from the representation learning task, allowing it to be pretrained independently before the initiation of the representation learning process. We also introduce a new strategy to encode the order of numbers and letters in each token of a time series, called spelling structure. This spelling structure is used to supervised the learned representations that are further used as conditions for a diffusion model. This supervision ensures that diffusion process does not overlook its condition that due to the cycle conditioning changes in its training iteration. Therefore, the network learn to effectively include the representation of conditioning samples in the reconstruction task of the diffusion model. Our strategy ensures that both the order of letters and the contextual relationships between tokens are effectively used in the representation learning process through self-supervised learning. As a result, the learned representations are versatile, making them applicable to any downstream task without concern that they are biased toward specific labels or annotations used during training.

While supervising the latent condition with spelling information aids in creating a multi-task representation, it has significant limitations due to the inherent sparsity of spelling information in categorical data. Each token's spelling is represented by an index, and the spelling information for an entire time series is encoded as a multi-hot vector with a dimensionality of $\mathcal{O}^n$, where $\mathcal{O}$ is the number of possible characters in the dataset, and $n$ is the length of the longest token. This results in a highly sparse representation. Consequently, using such a sparse vector to supervise the latent condition often produces a sparse latent representation, which is not ideal for capturing multi-task features and lacks the expressiveness needed to provide robust guidance for the diffusion process. To mitigate this issue, we propose to achieve this supervision through a learnable set of latent codes, where each code represents an initial direction toward a specific element of the spelling

---

[1]"tokens" refer to observations or their subcomponents, analogous to how large language models handle text.

vector. This setup enables the network to transfer its latent representation along $\mathcal{O}^n$ meaningful directions before supervising it with the sparse spelling vector. Since these initial directions are learnable and shared across all training samples, the network learns to explore meaningful semantic paths in the latent space, grouping time series that share token spelling similarities. For instance, two series $x = \{$'A01', 'B01'$\}$ and $y = \{$'B01', 'A02'$\}$ will be closer to each other than $x = \{$'A01', 'B01'$\}$ and $y = \{$'A01', 'B03'$\}$, despite the latter pair having more similarity in token order. This method is crucial for analyzing similarities between irregular time series, where data collection timelines are inconsistent, which is a common scenario in medical applications.

We focus on applying our method to ICD codes that represent medical trajectories and evaluate its performance on the widely recognized MIMIC-III and MIMIC-IV databases. This evaluation highlights the potential of our approach in advancing the understanding of medical data and its applications in clinical practice.

## 2 Related work

**Categorical data representation** Early research primarily focused on using metric learning to embed categorical data into a low-dimensional space (Jian et al., 2018; Li et al., 2023; Zhu et al., 2018; Jia et al., 2015). While demonstrating reasonable performance with a restricted and balanced number of tokens, their effectiveness diminishes in scenarios when there is a wide variation and imbalanced distribution of tokens. Following the successful introduction of mask modeling by BERT (Devlin et al., 2018), several algorithms (Yin et al., 2023; Majmundar et al., 2022; Du et al., 2023; Yan et al., 2024; Xie et al., 2024) have emerged aiming to take advantage of this model for representation learning of categorical data. While these methods perform well at capturing meaningful information, they often struggle with handling multidimensional time series data. In another line of research, authors proposed to utilize Graph Neural Networks for categorical representation learning (Liao & Li, 2023; You et al., 2020; Chen et al., 2024a) and demonstrated their effectiveness in capturing intricate dependencies of data. However, the computational expense of these networks hinders their scalability to large-scale datasets. Contrastive learning between different subsets of categories is another line of research that is considered for categorical data representation (Ucar et al., 2021). However, the performance of this method heavily depends on the split of the subsets, which is a hyperparameter, requiring additional effort for tuning.

**Diffusion models for categorical data** There are two approaches for adapting diffusion models to handle categorical data. In the first approach (Dieleman et al., 2022; Lin et al., 2023; Gao et al., 2022), input data is first converted into a continuous domain, and then a continuous diffusion model is employed to generate the data from a Gaussian distribution. These methods typically involve training a learnable embedding layer alongside the diffusion model. However, this setup leads to mutual dependence between the diffusion process and the embedding operation, leading to a suboptimal solution for both of the networks. Another direction of research explores a discrete diffusion process built on categorical distributions, thereby enabling their applicability for modeling inputs with binary elements(Hoogeboom et al., 2021; Austin et al., 2021; Savinov et al., 2021; Reid et al., 2022; Inoue et al., 2023). However, these methods have significant difficulties when employed in a conditional mode. This is because the condition, typically provided through a conditioning encoder, is continuous, whereas the diffusion process is discrete.

**Representation learning based on diffusion models** Learning representations through generative models has become a prominent focus in machine learning research. Much of this research revolves around Generative Adversarial Networks (GANs), (Radford et al., 2015; Chen et al., 2016) and Variational Autoencoders (VAEs) (Wang et al., 2023a), both of which benefit from a bottleneck that can be used as a representation layer. Recently, conditional diffusion models have demonstrated significant potential in discovering rich semantic information from visual data (Mittal et al., 2023; Preechakul et al., 2022; Zhang et al., 2022b; Yang & Wang, 2023; Wang et al., 2023b; Yue et al., 2024). Our work build on this recent advances in diffusion models for representation learning, notably extending this framework to handle categorical data. However, these methods commonly suffer from a significant drawback: they lack the ability to differentiate between the context that is intended to be contributed by noise and the one that needs to be contributed by the conditioning sample. Consequently, there is limited control over the meaning of the information captured by the representation layer. In contrast, our approach employs a self-supervised strategy that enables us to

enrich the representation with both contextual and semantic information of tokens, thereby enhancing the overall quality and interpretability of the learned representations.

## 3  Background

The fundamental concept of the Denoising Diffusion Model is to gradually introduce small amounts of noise to a clean input sample $S$, transforming it into a representation, $S_T$, that can be sampled from a Gaussian distribution. This transformation follows an incremental progression: $S_0 \to S_1 \to \cdots \to S_T$. The original data point $S_0$ is drawn from the data distribution $S_0 \sim q(S)$, while the conditional distribution $q(S_t \mid S_{t-1})$ is assumed to be Gaussian. Each noisy sample at time step $t$ is generated using the formula:

$$S_t = \sqrt{\beta_t} S_{t-1} + \sqrt{1-\beta_t} \epsilon_{t-1}, \tag{1}$$

where $\beta_t$ represents the noise schedule, and $\epsilon_{t-1}$ is sampled from a Gaussian distribution. By employing the notation $\alpha_t = 1 - \beta_t$ and $\bar{\alpha}_t = \prod_{i=1}^t \alpha_i$, $S_t$ can be sampled in closed form as

$$S_t = \sqrt{\bar{\alpha}_t} S_0 + \sqrt{1-\bar{\alpha}_t} \epsilon. \tag{2}$$

Thus, the true posterior of the sample can be represented as $q(S_{t-1} \mid S_t, S) = \mathcal{N}(S_{t-1}; \hat{\mu}_t, \hat{\beta}_t \mathbf{I})$, where $\hat{\mu}_t$ and $\hat{\beta}_t$ are computed using the following formulas:

$$\hat{\mu}_t := \frac{\sqrt{\bar{\alpha}_{t-1}} \beta_t}{1 - \bar{\alpha}_t} S + \frac{\sqrt{\alpha_t}(1 - \bar{\alpha}_{t-1})}{1 - \bar{\alpha}_t} S_t, \quad \hat{\beta}_t := \frac{1 - \bar{\alpha}_{t-1}}{1 - \bar{\alpha}_t} \beta_t. \tag{3}$$

In attempting to reverse this process, an approximate distribution is learned by $p_\theta(S_{t-1} \mid S_t) = \mathcal{N}(S_{t-1}; \mu_\theta(S_t, t), \hat{\beta}_t \mathbf{I})$, where $\mu_\theta$ is represented as

$$\mu_\theta = \frac{1}{\sqrt{\alpha_t}} \left( S_t - \frac{\beta_t}{\sqrt{1-\bar{\alpha}_t}} g_\theta(S_t, t) \right). \tag{4}$$

This representation allows the diffusion process to minimize the KL divergence between the posterior and the prior distribution of samples (Ho et al., 2020). $g_\theta(S_t, t)$ is a neural network that estimates the noise given the current state of the sample and the time step $t$. Therefore, sampling from $p_\theta(S_{t-1} \mid S_t)$ allows us to directly estimate $S_{t-1}$ as:

$$S_{t-1} = \frac{1}{\sqrt{\alpha_t}} \left( S_t - \frac{\beta_t}{\sqrt{1-\bar{\alpha}_t}} g_\theta(S_t, t) \right) + \frac{1 - \bar{\alpha}_{t-1}}{1 - \bar{\alpha}_t} \beta_t \epsilon. \tag{5}$$

This process enables the gradual denoising of a sample that was originally drawn from a noise distribution until it reaches a clean data sample from the distribution of clean samples.

It is important to note that the term "time step" is used in two distinct contexts in this work: in the definition of a categorical time series and in the operation of diffusion models. In the context of a time series, a "time step" refers to the specific time point at which each observation in the series is collected. In contrast, within diffusion models, a "time step" denotes a stage in the iterative process of adding noise to the input embedding. The number of time steps in a diffusion model controls the smoothness of the diffusion process, determining how progressively an input is transformed into its corresponding noise representation.

## 4  The Proposed Model

Given a time series comprising $r$ individual time steps, denoted as $w = \{w_1^T, w_2^T, \ldots, w_r^T\}^T$, where each time step includes a vector $w_i \in \mathbb{R}^{1 \times q_i}$, our method learns to convert this series into a meaningful embedding representation. Each $w_i$ contains a set of $q_i$ categorical values. Our approach consists of two main components: a data transformation module and a diffusion-based representation learning module. The data transformation serves as a learnable preprocessing step, performed independently before the initiation of the representation learning process.

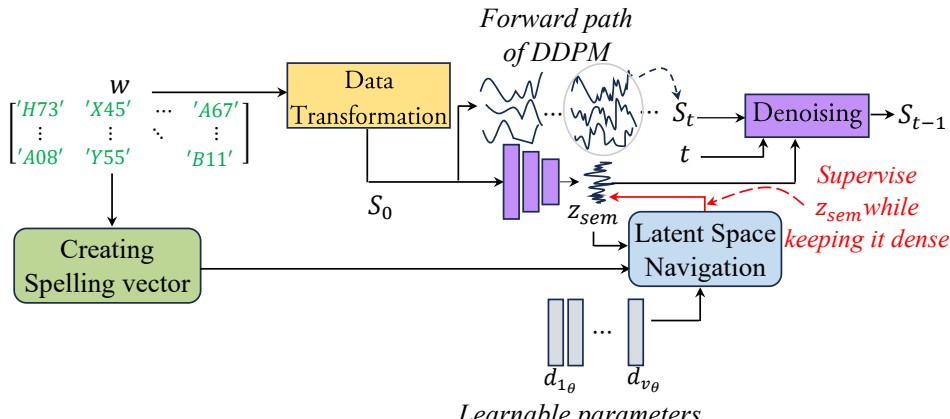

Figure 1: General pipeline of our method. The main objective is to extract a meaningful representation, $z_{sem}$, from a categorical time series $w$. The training process consists of two steps: first, transforming the categorical series into a continuous representation $S_0 = wA_{tr}$, where w is a multihor representation of $w$ and $A_{tr}$ is a linear transformation; second, projecting it into a latent space, $z_{sem}$, and utilizing it to guide a conditional diffusion model. $z_{sem}$ is supervised through Latent Space Navigation which allows this guidance to be dense enough while being supervised by a sparse vector. The sparse vector is provided by the Spelling information of the tokens in $w$ which is constructed using the methodology illustrated in Figure 12 (Appendix).

## 4.1 Data transformation

The embedding layer is a fundamental component in neural language modeling (Raffel et al., 2020; Liu et al., 2019; Devlin et al., 2018; Brown et al., 2020), functioning as a lookup table with learnable parameters. It associates each token within a text with a continuous representation, effectively converting each token into a fixed-size vector. However, this approach has limitations when dealing with categorical time series, where the number of tokens per time step varies and is not predefined. This variability can make it difficult to create consistent fixed-size vectors for different time steps. Furthermore, utilizing an embedding layer before a continuous diffusion model requires the network to optimize for two distinct, interdependent tasks. In this scenario, the diffusion model receives an input that changes with each training iteration because the input of the diffusion model is derived from the output of the embedding layer, which is itself optimized during training. As a result, noise in the diffusion process is added to an input that varies throughout the training process. Consequently, the network may not simulate a stable forward path during its denoising backward process, potentially leading to slow or even a lack of convergence.

To address this challenge, we propose an approach that learns to generate a fixed-size continuous representation "for each step" of a time series, irrespective of the number of tokens in each step. Similar to the embedding layers, we begin by establishing a vocabulary comprising all unique tokens in our dataset, assigning an index to each token. We then construct a binary vector for each time step of the series, where each element is set to one if there is a token in that time step whose vocabulary index matches the index of that element. This method allows us to create a multi-hot representation of the tokens present at each step of the time series. In contrast to traditional embedding layers, which typically position tokens close together based on their contextual co-occurrences, our approach emphasizes mapping one set of tokens close to another set when they differ by fewer than f tokens[2]. Each set represents all the tokens present in a single time step of a series. To facilitate this, we initialize a transformation function, $A_{tr}^{init}$ designed to map each multi-hot vector to a continuous code. One straightforward method to implement this transformation is by using a matrix that performs a linear projection on the input sample.

$$s_i = w_i A_{tr} \tag{6}$$

---

[2]In this work, f is empirically set to 5

where $w_i$ is the multihot representation of $w_i \in R^{1 \times q_i}$: $w_i = Multihot(w_i)$. For simplicity, we omit the explicit mention of subject indices in the notation of $w_i$. A subject index indicates the subject from which the $i$-th time step is selected. $A_{tr}$ is the transformation matrix which is initialized by random values $A_{tr}^{init} \in \mathbb{R}^{\mathcal{W} \times m}$, $w_i \in \mathbb{R}^{1 \times \mathcal{W}}$, $s_i \in \mathbb{R}^{1 \times m}$, $\mathcal{W}$ is the number of unique tokens in the vocabulary, and $m$ is the size of the continuous embedding vector. We normalize $w_i$ to have a unit $l_2$-norm before incorporating it into this equation, aiming to mitigate the influence of the number of tokens on the performance of our algorithm.

To ensure that contextually similar sets of tokens are mapped to nearby points, we first need to construct pairs of token sets that are contextually similar. For this purpose, we propose to initially introduce a set of minor alterations to $w_i$, by randomly changing some of its binary elements from ones to zeroes and vice versa. This approach ensures that the resulting set of tokens remains contextually similar to the original set, as most of the tokens are preserved. This resulting vector is denoted as $w_i^{new}$. Subsequently, we aim to find a new $A_{tr}$ that can map $w_i^{new}$ close to $s_i$, which has already been calculated as the embedding of $w_i$. To achieve this, we seek an $A_{tr}^{new}$ that satisfies the following equation.

$$y = w_i^{new} A_{tr}^{new}, \quad s.t. \; \|A_{tr}^{new}\|_2 = 1 \tag{7}$$

The constraint $\|A_{tr}^{new}\|_2 = 1$ is employed to normalize the transformation matrix, which prevents it from becoming too large or too small, thereby stabilizing the learning process. We define $y = \frac{s_i}{\|s_i\|}$. This normalization facilitates the mapping of $w_i^{new} A_{tr}^{new}$ and $s_i$ close to each other, while simultaneously avoiding a trivial solution for $A_{tr}^{new}$ that would transform all $w_i$ from different subjects to a fixed $s_i$, irrespective of their actual values.

To express this in matrix form, we concatenate all $w_i^{new}$ vertically to create a matrix: $W^{new} = \left[ (w_1^{new})^T, (w_2^{new})^T, \ldots, (w_{\mathcal{K}}^{new})^T \right]^T$, and similarly concatenate the different $s_i$ values to form $S = \left[ s_1^T, s_2^T, \ldots, s_{\mathcal{K}}^T \right]^T$. Then, $A_{tr}^{new}$ can be determined using the following equation:

$$A_{tr}^{new} = \left( (W^{new})^T W^{new} + \mu I \right)^{-1} (W^{new})^T S, \tag{8}$$

where $\mathcal{K}$ is the number of different $w_i$ considered for training $A_{tr}$. This approach results in a new transformation matrix $A_{tr}^{new}$ that learns to position contextually similar sets of tokens close to each other in the embedding space.

To further the training process, we utilize $A_{tr}^{new}$ to map the $w_i$ values into the embedding space, thereby achieving the corresponding $s_i$ values. Subsequently, we modify the $w_i$ values again and resolve equation (6) to obtain a new $A_{tr}^{new}$, repeating this process for a specified number of epochs. Algorithm 1 provides a summary of this training process for our data transformation strategy.

The algorithm is designed to take a single multi-hot vector, $w_i$, as input. Instead of using real data to create this vector, we can generate it efficiently by randomly placing ones in an all-zero vector. The number of ones corresponds to the average number of tokens that typically appear at each time step in our dataset. This approach allows us to simulate all possible token combinations without the need for additional data collection, making it a cost-effective way to cover a wide range of scenarios.

---

**Algorithm 1:** Transforming categorical data to continuous for time series analysis with multiple tokens per time step

---

**Input:** Multi-hot representation of tokens at each single time step, $w_i, i = 1, ..., \mathcal{K}$, Max epoch $\mathcal{Q}$
**Output:** The projection matrix $A_{tr}$ which transfers each $w_i$ to a continuous embedding vector $s_i$
1. Randomly initialize $A_{tr}$
2. $w_i^{l_2} = \frac{w_i}{\|w_i\|_2}, \quad \forall i \in [1, \mathcal{K}] \Rightarrow W \to W^{l_2}$
**for** $i \in [0, \mathcal{Q}]$ **do**
$\quad$ 3. Slight modification of $w_i, \quad \forall i \in [1, \mathcal{K}] \Rightarrow W \to W^{new}$
$\quad$ 4. $w_i^{new} = \frac{w_i^{new}}{\|w_i^{new}\|_2}, \quad \forall i \in [1, \mathcal{K}]$
$\quad$ 5. $y = W^{l_2} A_{tr}$
$\quad$ 6. $y_i = \frac{y_i}{\|y_i\|_2}, \quad \forall i \in [1, \mathcal{K}]$
$\quad$ 7. $A_{tr}^{new} = \left((W^{new})^T W^{new} + \mu I\right)^{-1} (W^{new})^T y$
$\quad$ 8. $A_{tr} = A_{tr}^{new}$
**end**

---

## 4.2 Diffusion based representation learning

The main objective of this module is to project a time series, represented in continuous form as $S_0 \in \mathbb{R}^{r \times m}$, into an embedding space to obtain a compact representation, $z_{sem} = E_\theta(S_0) \in \mathbb{R}^{m_z}$. To achieve this, we employ a conditional diffusion model trained in a self-supervised manner, where the condition and the target sample that the diffusion model learns to reconstruct are identical. However, the diffusion network is conditioned only on a compact representation of $S_0$, rather than on the entire time series. This encourages the network to capture as much relevant information as possible within this compact representation to enable accurate reconstruction of $S_0$. For simplicity in notation, we denote the learned representation as $z_{sem}$ which is then used as a condition for the diffusion model.

In practice, we sample a noisy version of the time series $S_0$ at a random time step $t$ as follows: $S_t = \sqrt{\bar{\alpha}_t} S_0 + \sqrt{1 - \bar{\alpha}_t} \epsilon$. We subsequently employ a denoising network, $g_\theta$, to provide an estimation of the noise in the noisy sample of $S_t$. With this value of noise, $g_\theta$ provides an estimation for the value of the noisy sample at time step $t - 1$:

$$S_{t-1} = \frac{1}{\sqrt{\alpha_t}} \Big( S_t - \frac{\beta_t}{\sqrt{1 - \bar{\alpha}_t}} g_\theta(S_t, t, z_{sem}) \Big) + \frac{1 - \bar{\alpha}_{t-1}}{1 - \bar{\alpha}_t} \beta_t \epsilon \tag{9}$$

Therefore, minimizing the difference between the posterior and prior distributions directly leads to parameter optimization for $E_\theta$, where $z_{sem} = E_\theta(S_0)$. This enables the diffusion process to extract meaningful information from $S_0$, as the learned features are crucial for guiding the diffusion process in distinguishing between the noise and the core content of the noisy sample.

### 4.2.1 Cycle conditioning

In this section, we introduce the concept of *cycle conditioned diffusion*, which aims to encourage our diffusion model to extract a multi-task representation from its conditioning samples. In traditional conditioning, diffusion models receive a clean version of a conditioning sample and use it to guide the denoising process toward the primary content of the conditioning sample. However, this method lacks a mechanism to ensure that the same latent representations are consistently extracted from contextually similar conditions that vary in fine details. As a result, conventional conditioning does not guarantee that the extracted representations focus exclusively on the main content of the conditioning samples and not on extraneous details.

To address this limitation, we propose incorporating *cycle consistency* in the feature extraction process from conditioning samples. By enforcing consistency in feature extraction, this approach ensures that the diffusion model learns to produce a multi-task representation from the conditioning samples. This method not only guides the diffusion process using the features of a clean condition but also applies the same guidance when the generated denoised sample is used as the condition for subsequent denoising steps. Consequently, we can predict an additional estimate for the mean of the prior distribution of samples.

$$\mu_\theta^{cycle} = \frac{1}{\sqrt{\alpha_t}}\Big(S_t - \frac{\beta_t}{\sqrt{1-\bar{\alpha}_t}}g_\theta(S_t, t, E_\theta(\frac{1}{\sqrt{\bar{\alpha}_t}}S_t - \frac{\sqrt{1-\bar{\alpha}_t}}{\sqrt{\bar{\alpha}_t}}\epsilon_{t-1})))\Big) \tag{10}$$

where $\epsilon_{t-1} = g_\theta(S_t, t, E_\theta(S_0))$. By enforcing this cycle constraint, the model learns to capture the same content from both the clean and the noisy samples. This prevents the model from simply distorting the latent condition to match the dimensions of $S_{t-1}$ and then using $S_t$ to add noise to this distorted sample. Instead, the model learns to perform the same guiding with different variations of the conditioning sample. Similar to vanilla DDPM, this new prior mean is drawn from minimizing KL divergence between the posterior distribtion of samples and the prior distribution using cycle consistent diffusion process.

### 4.2.2 Spelling supervision

One main disadvantage of cycle conditioning in diffusion models is the neglect of conditioning information. This occurs because the condition of each sample changes with every training iteration. As a result, the network cannot consistently use the conditioning information as guidance to reconstruct the sample. To prevent this issue, we propose supervising both the condition and the denoising process with additional information extracted from the tokens of the original time series before converting them to continuous form. This approach empowers the diffusion model to differentiate between the contributions of its conditioning information and that of the noisy sample, thereby preventing a blind fusion of these inputs, which could lead to an inaccurate feature representation of the conditioning sample.

To do so, we begin by constructing a spelling vector for each time series, which quantifies the spelling patterns of its tokens. Initially, we assign a numeric value, greater than 9, to each character in our database, i.e. assigning one number to each of the 26 letters in the English alphabet. Numeric values less than 10 are preserved to represent real numbers, enabling the encoding of tokens that contain both numerical and alphabetic characters. Each token is then represented as a numerical code by sequentially concatenating the assigned values for its characters, starting from the left. Zero-padding is applied if a token has fewer characters than the maximum length of tokens in our database. Subsequently, we create a zero-vector whose length matches the maximum possible numeric code in our dataset. Entries in this vector are set to one if there are tokens whose numeric code matches the corresponding index. This resulting vector is referred to as the spelling vector. This process is graphically depicted in the right panel of Figure 1.

For this supervision, we consider a function that receives the latent condition $z_{sem} = E_\theta(S_0)$ and learns to predict the elements of the spelling vector. Let $\rho = M_\theta(z_{sem})$ denote the predicted values of the spelling vector. The objective of the supervision is to minimize the cross-entropy error between the elements of the spelling vector $u$ and the predictions $\rho$.

$$L_{spl} = -\sum_x u_x log(\rho_x) \tag{11}$$

where $u_x$ is the $x$-th element of the spelling vector. To avoid the network from overlooking the condition, these spelling elements are additionally predicted when cycle conditioning the process, $\varpi_i = M_\theta(E_\theta(\frac{1}{\sqrt{\bar{\alpha}_t}}S_t - \frac{\sqrt{1-\bar{\alpha}_t}}{\sqrt{\bar{\alpha}_t}}\epsilon_{t-1}))$.

$$L_{cycle-spl} = -\sum_x u_x log(\varpi_x) \tag{12}$$

### 4.2.3 Latent Space Navigation

While spelling supervision prevents a denoising network from overlooking its condition, the effectiveness of this supervision is mostly hindered by the sparsity of information in the spelling vectors. Supervising the latent condition $z_{sem} = E_\theta(S_0)$ with a sparse vector yields a sparse latent condition, consequently reducing its effectiveness as a multi-task representation (Figure 2, right panel).

To avoid this issue, we propose to perform this supervision after navigating each latent condition in a set of meaningful directions (Figure 2, left panel). Our method involves learning a set of latent codes

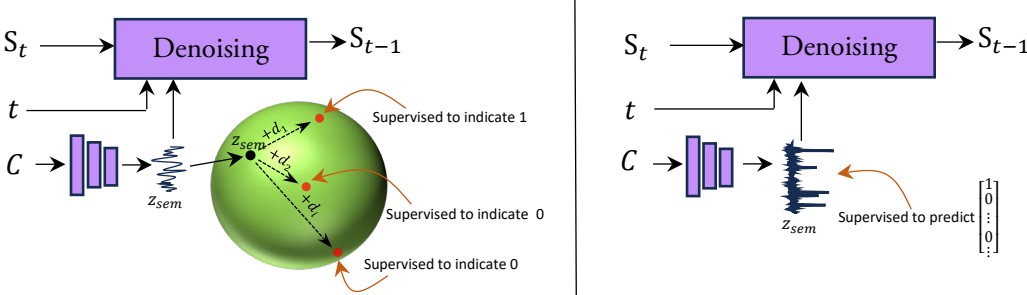

Figure 2: Direct supervision of the latent condition with spelling information from time series leads to a sparse latent space. This sparsity hinders its ability to serve as an effective vector for guiding the diffusion model or as a compact, multi-task representation (right panel). In contrast, Latent Space Navigation individually guides each component of the latent space by navigating it in a specific direction. This approach allows for a denser latent code, while supervised by the sparse spelling information from the time series (left panel).

$d_{x_\theta} \in \mathbb{R}^m, x \in \{1, ..., v\}^3$, each representing an initial direction towards one of the elements in the spelling vector, $u_x$. Consequently, $z_{sem} = E_\theta(S_0)$ is mapped to an individual target point in the latent space, $z_{sem \to u_x}$, before being employed to predict the $x$-th element of the spelling vector, $u_x$:

$$z_{sem \to u_x} = z_{sem} + d_{x_\theta} \tag{13}$$

This approach allows the network to avoid directly learning both the direction and magnitude of all $u_x$ values on the latent condition $E_\theta(S_0)$. Instead, the network learns the magnitude of each $u_x$ after traversing in the corresponding direction. This strategy facilitates the supervision of $z_{sem} = E_\theta(S_0)$ using a sparse spelling vector while ensuring sufficient density to function as an effective multi-task representation. The same navigational strategy is employed prior to predicting each spelling element $u_x$ from the cycle-conditioned model:

$$z_{csem \to u_x} = E_\theta \left( \frac{1}{\sqrt{\bar{\alpha}_t}} S_t - \frac{\sqrt{1 - \bar{\alpha}_t}}{\sqrt{\bar{\alpha}_t}} \epsilon_{t-1} \right) + d_{x_\theta} \tag{14}$$

where $\epsilon_{t-1} = g_\theta(S_t, t, E_\theta(S_0))$. Considering this navigation, the supervision terms in Equations (11) and (12) are reformulated as follows:

$$
\begin{aligned}
L_{spl} &= -\sum_x u_x log(M_\theta(z_{sem \to u_x})) \\
L_{cycle-spl} &= -\sum_x u_x log(M_\theta(z_{csem \to u_x}))
\end{aligned}
\tag{15}
$$

Detailed configurations of our method including the configuration of $E_\theta$ and $M_\theta$ are provided in Appendix 1.

### 4.2.4 Learning

We train our model in a self-supervised manner, where the condition and the input to the forward path of the diffusion model are identical. The overall loss function minimizes the MSE loss between the predicted and the true noise, considering both the base and the cycle conditioning of the denoising network. Additionally, it includes the prediction loss for spelling errors associated with these two conditioning mechanisms.

In practice, we leverage a Multi-Step Optimization approach, which allows the introduction of extra loss functions alongside the MSE loss of the diffusion models with minimal impact on their performance. Specifically, we alternate between optimizing the MSE and cross-entropy losses. The cross-entropy losses, $L_{spl}$ and $L_{cycle-spl}$, are only applied if any of the MSE losses exceed a threshold of 0.05. During the multi-step optimization, we

---

[3]$v = \mathcal{O}^n$ where $\mathcal{O}$ is the number of possible characters (letters or numbers) in the dataset, and $n$ is the length of the longest token

perform a single forward pass to compute both the DDPM's MSE losses and the cross-entropy losses. In the first step, we update the model parameters with respect to the MSE losses while freezing the components $M_\theta$ and $d_\theta$ affected by the cross-entropy losses. In the next step, we update the model with respect to the cross-entropy losses, freezing the components relevant to the MSE losses.

### 4.2.5 Inference

Inference strategies may vary depending on the downstream task. When a substantial amount of training data with corresponding labels is available, we can directly extract representations from the latent space of the conditioning encoder, $E_\theta(S_0)$. These representations are sufficient for effectively training a downstream classifier.

Conversely, in scenarios with a limited number of training samples, it is advantageous to leverage the generative capabilities of our method. In this case, representations are derived from $E_\theta(S_t, t, E_\theta(\frac{1}{\sqrt{\bar{\alpha}_t}} S_t - \frac{\sqrt{1-\bar{\alpha}_t}}{\sqrt{\bar{\alpha}_t}} \epsilon_{t-1}))$, where $S_t = \sqrt{\bar{\alpha}_t} S_0 + \sqrt{1 - \bar{\alpha}_t}\epsilon$, and $\epsilon$ is randomly sampled from a Gaussian distribution.

## 5 Experiments

Our experiments are conducted using two well-known databases, MIMIC III (Johnson et al., 2016) and MIMIC IV (Johnson et al., 2023), each containing records of medical activities from various patient visits to a hospital. We construct a time series for each patient in these databases. These series are built from the diagnosis codes of the patients, as they have minimal missing values compared to other activity records in these databases. The diagnosis codes collected during each patient's visit, form a feature vector, representing one step of a time series. Each feature vector is truncated to a length of 13, enhancing the computational efficiency of our analysis. We only consider the first four visits of each patient. While MIMIC IV contains both ICD-9 and ICD-10 codes, MIMIC III only includes ICD-9 codes. To ensure consistency, we truncate the codes to their first three characters and prefix them with either '9' or '10' to denote their coding system. This ensures the maintenance of relevant information while excluding unnecessary details from our analysis.

**Data split**: Most of our experiments are conducted on MIMIC IV, where we randomly select 126,736 time series for training our representation learning method. The evaluation sets for different tasks are composed of another 31,684 randomly selected samples, which are used to evaluate the performance of the learned representations on various downstream tasks. For each task, positive cases include patients diagnosed with a specific disease, while negative samples are randomly selected from the remaining patients, ensuring an equal number to the positive cases. Therefore, not all samples are used for training the downstream classifiers. For MIMIC III, all samples serve as training data for the representation learning model. The training and the evaluation of the downstream task are then conducted using the same split of the MIMIC IV.

**Preparing data for the downstream task**: After training our diffusion model, the training and test samples are separately projected into the embedding layer of this representation learning framework. The representations of the training samples are then utilized to train a downstream classifier. This classifier learns to predict a specific disease within a time window of 4 months before it is diagnosed by physicians in the real world. Patients who are diagnosed with the disease before this specified time window are excluded from the training process of the downstream classifier. For the downstream classifier, we employed Support Vector Machines (Cortes & Vapnik, 1995), Gaussian Process (Williams & Rasmussen, 2006), Random Forest (Breiman, 2001), Adaboost (Freund et al., 1999), and Gaussian Naive Bayes (GaussianNB), which are selected based on their proven performance in similar tasks.

**Evaluation metrics** For evaluation, we use the following metrics: AUROC: This measures the ability of a classifier to distinguish between positive and negative classes across different threshold values. AUPRC: This represents the trade-off between Precision and Recall of a classifier, indicating its ability to correctly classify positive samples as positive. Cohen's Kappa: This measures how often classifiers correctly classify positive cases, while also considering that some correct predictions might occur by chance. Macro-average F1-score: This is an average F1 score for each of the positive and negative classes, treating them equally regardless of the number of positive and negative cases among the evaluation samples.

| | Our method | | SwitchTab (Wu et al., 2024) | | ReconTab (Chen et al., 2023) | | SCARF (Bahri et al., 2021) | | Em-MLM (Devlin et al., 2018) | | OutBERT-MLM (Devlin et al., 2018) | | Em-AuCon (Chen et al., 2024b) | |
|---|---|---|---|---|---|---|---|---|---|---|---|---|---|---|
| **Heart failure prediction** | | | | | | | | | | | | | | |
| | AUROC | AUCPR | AUROC | AUCPR | AUROC | AUCPR | AUROC | AUCPR | AUROC | AUCPR | AUROC | AUCPR | AUROC | AUCPR |
| Support Vector | $91.81_{\pm0.61}$ | $12.49_{\pm5\%}$ | $90.39_{\pm0.76}$ | $09.57_{\pm6\%}$ | $61.79_{\pm0.49}$ | $02.54_{\pm5\%}$ | $66.39_{\pm0.85}$ | $06.48_{\pm7\%}$ | $66.49_{\pm0.44}$ | $03.15_{\pm4\%}$ | $74.34_{\pm0.52}$ | $04.46_{\pm8\%}$ | $77.00_{\pm0.84}$ | $05.11_{\pm9\%}$ |
| Gaussian Process | $90.96_{\pm0.62}$ | $13.70_{\pm5\%}$ | $90.23_{\pm0.47}$ | $10.09_{\pm4\%}$ | $87.57_{\pm0.68}$ | $09.86_{\pm6\%}$ | $79.85_{\pm0.36}$ | $06.13_{\pm4\%}$ | $64.10_{\pm0.45}$ | $02.86_{\pm4\%}$ | $76.63_{\pm0.56}$ | $05.01_{\pm5\%}$ | $76.22_{\pm0.69}$ | $07.69_{\pm7\%}$ |
| Random Forest | $89.48_{\pm0.37}$ | $14.21_{\pm5\%}$ | $86.12_{\pm0.38}$ | $07.47_{\pm4\%}$ | $84.04_{\pm0.62}$ | $06.72_{\pm5\%}$ | $80.87_{\pm0.67}$ | $06.46_{\pm5\%}$ | $67.63_{\pm0.44}$ | $03.31_{\pm4\%}$ | $68.69_{\pm0.62}$ | $03.48_{\pm6\%}$ | $77.33_{\pm0.74}$ | $05.26_{\pm5\%}$ |
| Ada Boost | $85.12_{\pm0.73}$ | $08.62_{\pm6\%}$ | $80.53_{\pm0.66}$ | $05.54_{\pm6\%}$ | $65.87_{\pm0.37}$ | $03.04_{\pm4\%}$ | $78.04_{\pm0.41}$ | $05.56_{\pm5\%}$ | $64.28_{\pm0.47}$ | $02.88_{\pm6\%}$ | $66.62_{\pm0.65}$ | $03.17_{\pm6\%}$ | $71.21_{\pm0.40}$ | $03.87_{\pm5\%}$ |
| GaussianNB | $89.20_{\pm0.76}$ | $08.41_{\pm7\%}$ | $80.16_{\pm0.63}$ | $05.28_{\pm5\%}$ | $64.13_{\pm0.55}$ | $02.79_{\pm4\%}$ | $77.93_{\pm0.62}$ | $05.46_{\pm5\%}$ | $60.58_{\pm0.68}$ | $02.54_{\pm6\%}$ | $70.04_{\pm0.74}$ | $03.63_{\pm5\%}$ | $74.95_{\pm0.63}$ | $04.67_{\pm7\%}$ |
| | Av.Kappa | Av.BF1 | Av.Kappa | Av.BF1 | Av.Kappa | Av.BF1 | Av.Kappa | Av.BF1 | Av.Kappa | Av.BF1 | Av.Kappa | Av.BF1 | Av.Kappa | Av.BF1 |
| | $18.84_{\pm0.69}$ | $52.26_{\pm0.73}$ | $17.09_{\pm0.48}$ | $50.42_{\pm0.64}$ | $12.72_{\pm1.14}$ | $43.15_{\pm0.59}$ | $16.04_{\pm0.77}$ | $49.41_{\pm0.48}$ | $14.22_{\pm0.67}$ | $45.13_{\pm0.83}$ | $15.74_{\pm0.54}$ | $48.11_{\pm0.83}$ | $16.09_{\pm0.82}$ | $48.92_{\pm0.74}$ |
| **Lung cancer prediction** | | | | | | | | | | | | | | |
| Support Vector | $90.51_{\pm0.64}$ | $0.54_{\pm9\%}$ | $87.98_{\pm0.56}$ | $0.42_{\pm8\%}$ | $60.25_{\pm0.72}$ | $0.15_{\pm8\%}$ | $86.75_{\pm0.54}$ | $0.46_{\pm12\%}$ | $83.47_{\pm0.77}$ | $0.34_{\pm9\%}$ | $82.61_{\pm0.44}$ | $0.42_{\pm6\%}$ | $80.44_{\pm0.58}$ | $0.30_{\pm8\%}$ |
| Gaussian Process | $88.79_{\pm0.86}$ | $0.56_{\pm8\%}$ | $86.26_{\pm0.74}$ | $0.43_{\pm9\%}$ | $63.71_{\pm0.72}$ | $0.18_{\pm7\%}$ | $85.18_{\pm0.61}$ | $0.38_{\pm6\%}$ | $80.14_{\pm0.37}$ | $0.31_{\pm4\%}$ | $82.94_{\pm0.67}$ | $0.37_{\pm7\%}$ | $80.01_{\pm0.39}$ | $0.31_{\pm6\%}$ |
| Random Forest | $91.35_{\pm0.48}$ | $0.59_{\pm5\%}$ | $85.91_{\pm0.63}$ | $0.36_{\pm5\%}$ | $87.39_{\pm0.56}$ | $0.34_{\pm5\%}$ | $82.82_{\pm0.39}$ | $0.37_{\pm4\%}$ | $84.19_{\pm0.61}$ | $0.38_{\pm6\%}$ | $80.14_{\pm0.44}$ | $0.33_{\pm5\%}$ | $82.19_{\pm0.83}$ | $0.35_{\pm12\%}$ |
| Ada Boost | $86.60_{\pm0.57}$ | $0.49_{\pm6\%}$ | $82.12_{\pm0.49}$ | $0.31_{\pm4\%}$ | $59.49_{\pm0.73}$ | $0.13_{\pm8\%}$ | $85.57_{\pm0.76}$ | $0.36_{\pm7\%}$ | $81.25_{\pm0.43}$ | $0.24_{\pm4\%}$ | $83.81_{\pm0.62}$ | $0.31_{\pm5\%}$ | $77.19_{\pm0.60}$ | $0.22_{\pm7\%}$ |
| GaussianNB | $84.39_{\pm0.77}$ | $0.43_{\pm6\%}$ | $88.32_{\pm0.45}$ | $0.44_{\pm5\%}$ | $62.35_{\pm0.65}$ | $0.14_{\pm8\%}$ | $84.33_{\pm0.67}$ | $0.33_{\pm5\%}$ | $82.12_{\pm0.38}$ | $0.29_{\pm4\%}$ | $80.14_{\pm0.73}$ | $0.24_{\pm6\%}$ | $81.17_{\pm0.52}$ | $0.36_{\pm4\%}$ |
| | Av.Kappa | Av.BF1 | Av.Kappa | Av.BF1 | Av.Kappa | Av.BF1 | Av.Kappa | Av.BF1 | Av.Kappa | Av.BF1 | Av.Kappa | Av.BF1 | Av.Kappa | Av.BF1 |
| | $0.94_{\pm9\%}$ | $45.87_{\pm0.73}$ | $0.85_{\pm0.8\%}$ | $43.33_{\pm0.64}$ | $0.33_{\pm14\%}$ | $41.59_{\pm0.78}$ | $0.70_{\pm7\%}$ | $49.41_{\pm0.48}$ | $0.23_{\pm13\%}$ | $40.66_{\pm0.72}$ | $0.77_{\pm8\%}$ | $43.44_{\pm0.67}$ | $0.64_{\pm0.66}$ | $44.19_{\pm9\%}$ |

Table 1: Performance comparison between our method and the state-of-the-art on heart failure and lung cancer prediction

## 5.1 Performance Evaluation in Healthcare Applications

In this section, we evaluate the performance of our diffusion-based method in learning representations that are effective in various downstream tasks. Specifically, we consider heart failure and lung cancer as two distinct tasks that are completely unrelated and need highly expressive representations for accurate identification. Therefore, demonstrating the capability of a single representation in predicting both of these tasks can prove the multi-task utility of that representation. Our evaluation includes comparisons with the state-of-the-art methods in representation learning from categorical data: SwitchTab(Wu et al., 2024), ReConTab (Chen et al., 2023), and SCARF (Bahri et al., 2021).

Additionally, we compare our results with the representations learned by *the embedding layer* and *one layer before the last* of the BERT model (Devlin et al., 2018), pretrained using Mask Modeling. These methods are respectively denoted as Em-MLM and OutBERT-MLM. Furthermore, we compare our method with the same BERT structure, trained using the self-supervised learning strategy proposed by Chen et al. (Chen et al., 2024b), referred to as Em-AuCon. For each method, we consider a four-month window preceding disease diagnosis, providing an optimal timeframe for initiating medication upon correct disease prediction, thus reducing the risk of fatality associated with these diseases. The representation dimension of our method, along with the models that utilize BERT as their backbone architecture, are set to 400. For SwitchTab(Wu et al., 2024), ReConTab (Chen et al., 2023), and SCARF (Bahri et al., 2021), we follow the same protocol as described in their original papers, resulting in representation dimensions of 52, 256, and 256, respectively.

As can be observed, our method outperforms all other strategies across all the metrics. For instance, in heart failure prediction, it surpasses SwitchTab's best results by 1.42%, which itself is the leading performance among the competing strategies. Even the lowest performance of our method, achieved by Adaboost, is 94.13% of the best results achieved by other competitors, demonstrating consistent performance of our method despite using different classifiers for the downstream task. In the case of the lung cancer prediction, our best result outperforms the best results of the other competitors by 3.3%, while the lowest performance of our method still achieves 95.57% of this best result.

## 5.2 Probabilistic representation learning

In this section, we demonstrate the effectiveness of our method in generating different meaningful representations for a single input sample. This capability is particularly valuable in applications where there is a limited number of labeled samples to train a downstream classifier.

In this case, instead of directly obtaining representations from $E_\theta(S_0)$, we acquire the representations from $E_\theta(\frac{1}{\sqrt{\beta_t}}S_t - \frac{\sqrt{1-\beta_t}}{\sqrt{\beta_t}}\epsilon_{t-1})$, where $\epsilon_{t-1}$ is the estimated noise of the noisy sample $S_t$, estimated by $g_\theta(.)$. For

this, we consider $\mathcal{B}$ random $S_t$ with $\beta_t$ value between $1e^{-4}$ and $0.5e^{-2}$ for each positive patient to generate various representations of the disease. Notably, only one forward pass is needed to compute each $e_{t-1}$ from $S_t$ with no iterative process involved, making our method exceptionally fast in generating novel representations of samples. By employing this simple strategy, we ensure there is enough number of training samples for the downstream classifiers, which is $\mathcal{B} =15$ or 30 times more training samples compared to our baseline model.

|  | Our method (Prob. #15) | | Our method (Prob. #30) | |
|---|---|---|---|---|
|  | AUROC | AUCPR | AUROC | AUCPR |
| Support Vector | 92.38±0.63 | 16.92±5% | 93.80±0.53 | 19.14±5% |
| Gaussian Process | 92.03±0.46 | 17.00±4% | 92.96±0.37 | 17.93±4% |
| Random Forest | 91.84±0.54 | 16.14±6% | 92.33±0.65 | 16.32±5% |
| Ada Boost | 87.91±0.73 | 11.53±6% | 88.76±0.55 | 11.96±5% |
| GaussianNB | 89.87±0.64 | 12.75±6% | 90.32±0.53 | 14.17±4% |

Table 2: Probabilistic representation learning using our method for heart failure prediction. The number of synthesized samples for each real patient is 15 and 30, respectively.

|  | Our method (Prob. #30) | | Our method (Prob. #60) | |
|---|---|---|---|---|
|  | AUROC | AUCPR | AUROC | AUCPR |
| Support Vector | 93.17±0.43 | 0.74±4% | 93.73±0.63 | 0.81±4% |
| Gaussian Process | 90.82±0.57 | 0.55±5% | 90.94±0.35 | 0.56±5% |
| Random Forest | 92.68±0.72 | 0.70±7% | 92.85±0.62 | 0.71±5%− |
| Ada Boost | 89.95±0.44 | 0.51±4% | 90.35±0.47 | 0.53±4% |
| GaussianNB | 89.83±0.51 | 0.50±5% | 90.10±0.63 | 0.51±6% |

Table 3: Probabilistic representation learning using our method for lung cancer prediction. The number of synthesized samples for each real patient is 15 and 30, respectively.

**Results**: As demonstrated in Tables 2 and 3, our strategy clearly outperforms the baseline model in both the downstream tasks. For instance, we observe an overall increase in AUROC for both the tasks. Specifically, the performance of our method improves from 91.81% to 92.38% for heart failure prediction and from 91.35% to 93.17% for lung cancer prediction, with just 15 additional representations for each positive sample. In contrast, achieving AUROC scores of 93.80% and 93.73% for these tasks requires 30 times more training samples for the downstream classifier.

### 5.3 Our data transformation strategy vs Embedding layers

In this section, we evaluate the ability of Algorithm 1 to capture meaningful information from a sequence of categorical tokens, comparing its performance to that of the BERT embedding layer. Following strategies from prior works (Kennard et al., 2016; Bojanowski et al., 2017; Pennington et al., 2014), which propose to evaluate the embedding layers based on their capacity to preserve word similarities in the embedding space, we examine how effectively each embedding layer maintains sentence-level similarity within its embedding layer. This property is especially relevant when outputs of the embedding layers are further used as inputs to diffusion models, where each sample is treated as a distinct point in a high-dimensional space rather than as a composite of multiple subspaces.

To evaluate this capability, we first concatenate the textual descriptions of all ICD codes associated with each visit in every time series within our database. We then randomly select two visits from different series and use a pre-trained large language model to obtain embeddings for these concatenated descriptions. By calculating a cosine similarity score between these embeddings, we measure the ability of each embedding layer to capture the semantic similarity between two distinct sets of ICD codes. For evaluation, we transform each set of ICD codes (corresponding to individual patient visits) into a continuous representation using both our method and the BERT embedding layer. We then calculate a cosine similarity score for each embedding, which enables us to assess how effective each method correlates with the similarity scores derived from the ICD code descriptions. We adopt two distinct strategies for learning parameters in a BERT embedding layer. The first approach is through Masked Language Modeling (MLM), a conventional method commonly utilized for self-supervised training of the embedding layers in large language models.

Additionally, we explore an extension of the method proposed by (Chen et al., 2024b), which is a multi-task strategy for self-supervised representation learning from images. However, we adapted this strategy for categorical data to enable the capture of relationships between different tokens in the immediate output of the embedding layer. In the case of masked modeling, we extract the embedding data from two different layers: one from the conventional embedding layer and another from the layer preceding the softmax operation at the end of the network. This is because masked modeling does not inherently encourage the embedding layer to accumulate the contextual relationships between different words in each embedded token. Instead, it captures these relationships at the end of the network, after the transformers of BERT, where it needs to estimate a masked token from a set of surrounding tokens, and therefore needs to accumulate this surrounding information for each token.

Table 4 presents a comparison between these four strategies using Spearman correlation. Our method shows a high positive correlation across different embedding dimensions, suggesting a strong association between its similarity scores and the ground truth similarities. In contrast, the BERT embedding layer trained by MLM returns the lowest correlation, aligning with our premise that masked modeling does not inherently capture the contextual relationship between different tokens.

| | Dimensionality | | | | | | | |
| | 40 | 100 | 200 | 300 | 400 | 700 | 1000 | 1500 |
|---|---|---|---|---|---|---|---|---|
| Em-MLM | $0.14{\pm}0.11$ | $0.17{\pm}0.06$ | $0.18{\pm}0.13$ | $0.18{\pm}0.07$ | $0.15{\pm}0.07$ | $0.17{\pm}0.06$ | $0.19{\pm}0.04$ | $0.20{\pm}0.08$ |
| Out-MLM | $0.33{\pm}0.08$ | $0.34{\pm}0.08$ | $0.34{\pm}0,.08$ | $0.36{\pm}0.10$ | $0.36{\pm}0.08$ | $0.37{\pm}0.05$ | $0.38{\pm}0.07$ | $0.39{\pm}0.07$ |
| Em-ConAu | $0.18{\pm}0.07$ | $0.21{\pm}0.12$ | $0.22{\pm}0.12$ | $0.23{\pm}0.05$ | $0.24{\pm}0.10$ | $0.27{\pm}0.03$ | $0.28{\pm}0.06$ | $0.29{\pm}0.06$ |
| Out-ConAu | $0.38{\pm}0.12$ | $0.39{\pm}0.11$ | $0.39{\pm}0.09$ | $0.41{\pm}0.08$ | $0.42{\pm}0.09$ | $0.44{\pm}0.08$ | $0.46{\pm}0.09$ | $0.45{\pm}0.05$ |
| Algorithm 1 | $0.42{\pm}0.11$ | $0.43{\pm}0.07$ | $0.44{\pm}0.09$ | $0.45{\pm}0.06$ | $0.45{\pm}0.07$ | $0.49{\pm}0.08$ | $0.51{\pm}0.06$ | $0.51{\pm}0.07$ |

Table 4: Performance comparison between our method and the BERT embedding layer on the MIMIC$-$IV dataset based on Spearman correlation over 20 runs.

### 5.4   Ablation study

We also conducted an ablation analysis focused on answering the following key questions:

1. **Does cycle conditioning aid in extracting more versatile representations?** To investigate this, we trained two different versions of our method: one incorporating cycle conditioning loss and the other without it, while preserving the spelling loss functions in both versions. This design allows us to evaluate the specific contribution of the cycle conditioning loss to the overall performance of our method. As shown in Table 5, the network without cycle conditioning achieves significantly lower accuracy in the heart failure prediction task, demonstrating the effectiveness of this loss function in enhancing the performance of our method.

2. **Is spelling supervision necessary?** The conditioning loss function may encourage the network to extract the same representation from both a clear conditioning sample and its noisy version, suggesting it could be sufficient for effective representation learning. To explore this, we trained our model using the cycle conditioning loss but without the spelling loss functions. To prevent the network from extracting identical representations from all conditioning samples, we normalized this layer to have the same standard deviation as the input conditioning sample. The results of this comparison are also presented in Table 5. As observed, without the spelling loss function, our model struggles to capture meaningful representations from the conditioning sample, which hinders its ability to effectively estimate heart failure as the downstream task.

| | Full model | | w/o cycle conditioning | | w/o spelling supervision | |
| | AUROC | AUCPR | AUROC | AUCPR | AUROC | AUCPR |
|---|---|---|---|---|---|---|
| Support Vector | $91.81{\pm}0.61$ | $12.49{\pm}5\%$ | $89.47{\pm}0.44$ | $12.23{\pm}7\%$ | $63.21{\pm}0.71$ | $02.78{\pm}9\%$ |
| Gaussian Process | $90.96{\pm}0.62$ | $13.70{\pm}5\%$ | $87.93{\pm}0.72$ | $09.02{\pm}9\%$ | $62.16{\pm}0.54$ | $02.14{\pm}9\%$ |
| Random Forest | $89.48{\pm}0.37$ | $14.21{\pm}5\%$ | $88.51{\pm}0.61$ | $12.61{\pm}7\%$ | $64.21{\pm}0.47$ | $02.84{\pm}5\%$ |
| Ada Boost | $85.12{\pm}0.73$ | $08.62{\pm}6\%$ | $83.77{\pm}0.67$ | $07.99{\pm}6\%$ | $58.12{\pm}0.43$ | $02.11{\pm}11\%$ |
| GaussianNB | $89.20{\pm}0.76$ | $08.41{\pm}7\%$ | $87.23{\pm}0.58$ | $08.09{\pm}4\%$ | $58.69{\pm}0.84$ | $02.26{\pm}8\%$ |

Table 5: Ablation study on our proposed method where the effectiveness of its two main components, cycle conditioning and spelling supervision are evaluated

## 6   Conclusion

In conclusion, we introduced a novel diffusion-based method for representation learning in categorical data. We proposed that cycle conditioning of diffusion models can effectively extract meaningful and versatile

information from the conditioning samples. Additionally, we introduced a novel strategy to supervise the conditioning process with extra information about the spelling of tokens, thus preventing the overlooking of the conditioning samples in the denoising process. To enhance the robustness of the representations, we proposed supervising the latent condition through linear navigation in its latent space. We conducted a comprehensive set of experiments to demonstrate the effectiveness of our diffusion model in extracting meaningful representations from categorical data. However, as with any approach, this method presents certain limitations, including significant computational demands inherent to diffusion-based models. Future research may focus on improving computational efficiency, potentially by integrating faster generative modeling techniques for representation learning tasks. Furthermore, evaluating the model's performance on a wider range of databases would offer valuable insights into its robustness and adaptability. Specifically, exploring its sensitivity to diverse categorical datasets, beyond the medical domain, could provide a deeper understanding of its capability to learn representations in different contexts and inform its application to medical time series representation learning.

## Acknowledgements

This study was part of the AIR Lund (Artificially Intelligent Use of Registers at Lund University) research environment and received funding from the Swedish Research Council (VR; grant no. 2019-00198). It also received funding from the Knowledge Foundation (CAISR Health with grant 20200208 01H). The funders had no role in study design, data collection and analysis, decision to publish, or preparation of the manuscript.

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

# A  Implementation details of the BERT embedding layers

In our study, we utilize BERT as the foundational architecture to evaluate the performance of embedding layers. BERT's embedding layer is initially designed for neural language modeling, which typically requires a large token vocabulary. To adapt it for our medical application, which has a smaller vocabulary, we introduce a fully connected layer before the embedding layer. This modification allows us to control the size of the embedding space and identify the most suitable dimensions for our application. Additionally, we add a linear layer at the end of the network to generate a logits vector for each index in the output sequence. The length of this logits vector corresponds to the total number of tokens in our medical vocabulary. We follow the standard masked modeling procedure by randomly masking 15% of the ICD codes, where 80% of the masked codes are replaced with a predefined token, 10% are substituted with other ICD codes, and 10

We use the same training dataset as for our representation learning model. The learning rate is set to 1e-5, we employ the AdamW optimizer, and use cross-entropy loss to assess the estimated indices for the masked tokens.

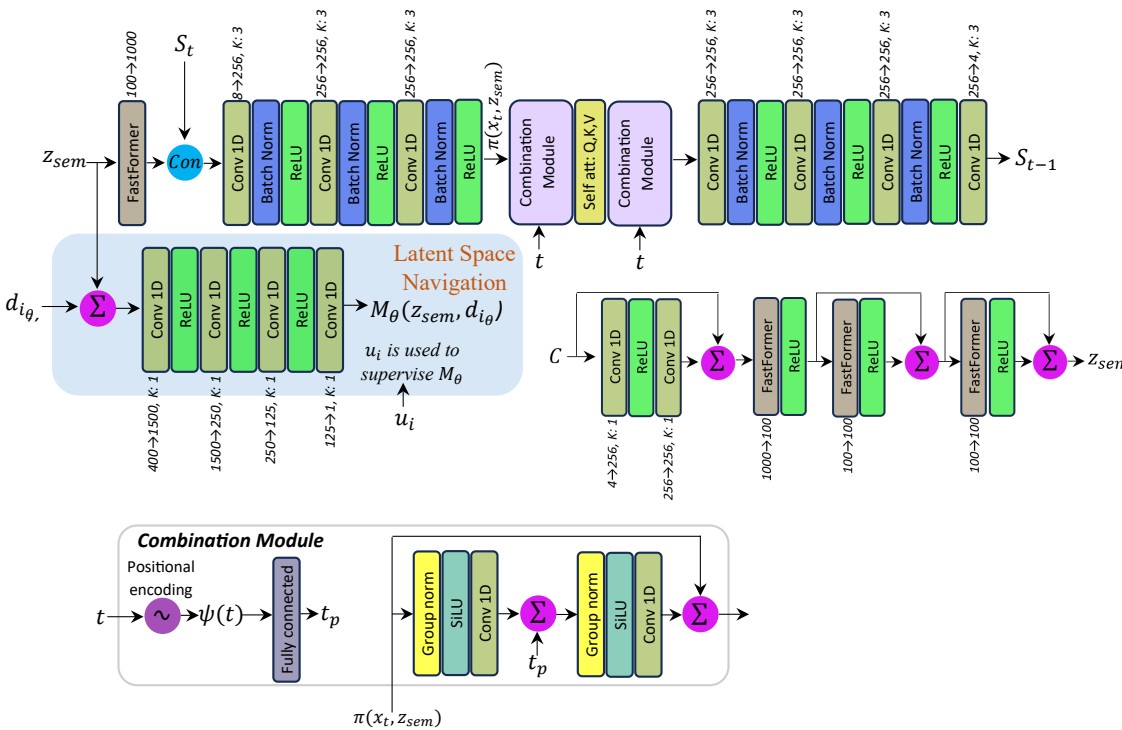

Figure 3: Implementation details of our method: For simplicity, we avoid illustrating the cycle conditioning part, which necessitates reusing $S_{t-1}$ as the conditioning sample instead of $C$.

# B  Our method vs other diffusion based strategies

We conduct an additional experiment to evaluate the effectiveness of our method in comparison to other strategies that utilize diffusion modeling for representation learning. Specifically, we compare our approach with those presented in (Mittal et al., 2023; Preechakul et al., 2022; Wang et al., 2023b). It is important to note that we do not include comparisons with (Zhang et al., 2022b; Yang & Wang, 2023), as their methods focus on extracting representations from a pre-trained diffusion model without using a conditioning mechanism.

**Fair comparison**: To ensure a fair comparison, we use the same structure for the conditioning encoder and denoising components in all the diffusion models. The only variations are in the transformation of categorical features to continuous ones and in the formulation of loss functions, where different supervision methods are

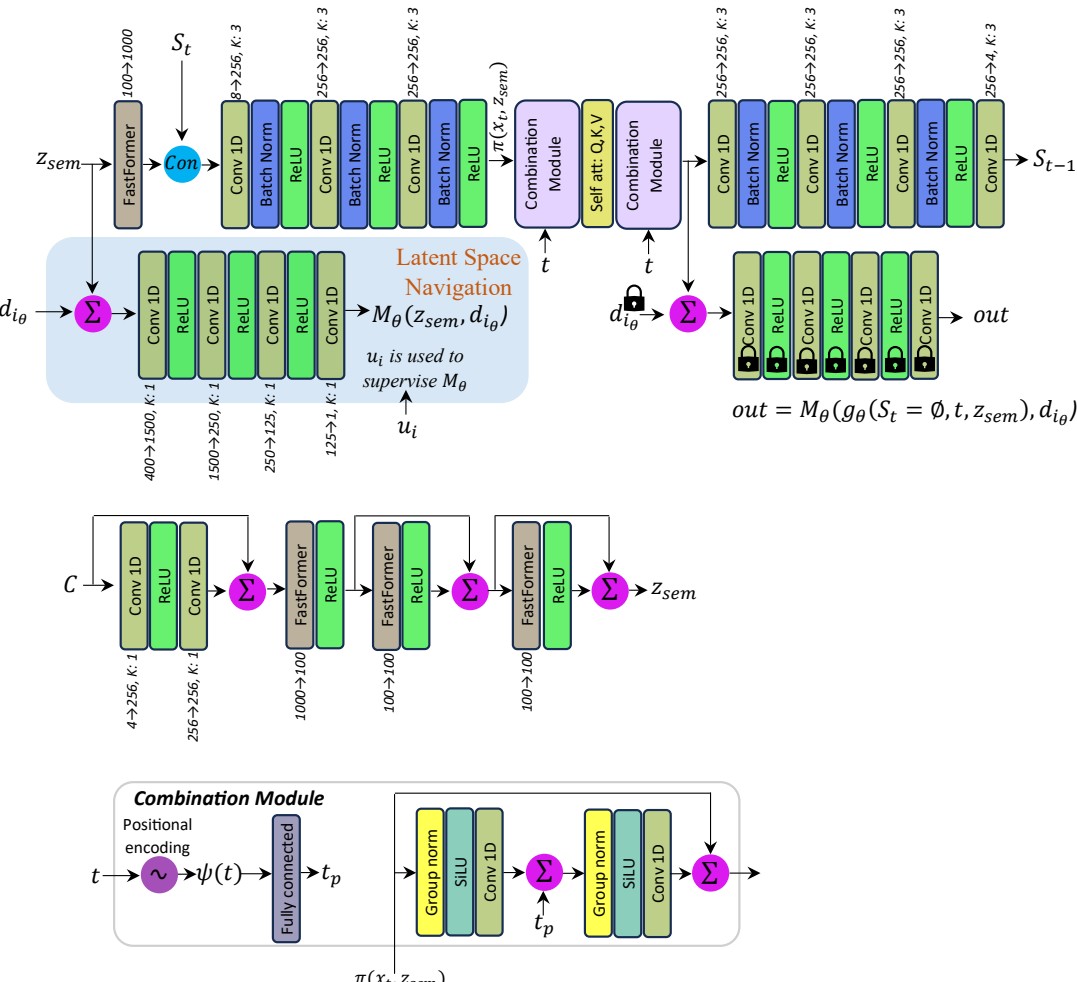

Figure 4: Implementation details of our method when cycle conditioning is replaced with the disentangled guidance strategy proposed by (Bhunia et al., 2023).

applied to the embedding layers. A detailed description of the configurations and associated loss functions can be found in Figures 3-11. Here, we denote the conditioning sample as $C$ and the input sample at time step $t$ as $S_t$ to distinguish between them.

**Results**: As shown in Table 6, our method outperforms all competing strategies. It is important to note that none of these methods were originally designed for learning representations from categorical samples, necessitating the inclusion of additional strategies to adapt them for categorical data. We employed the approach outlined in (Lin et al., 2023; Dieleman et al., 2022), which involves making an extra prediction on the embedding layer before it is fed into the diffusion model.

Additionally, we conducted an experiment where the same embedding input is used at the end of the network to maintain consistency in the intermediate representations, as illustrated in Figure 9. However, our experiments revealed that this approach encounters significant convergence issues, primarily due to the mutual dependence between the embedding layer and the diffusion model.

For each method, we examine two different configurations, denoted by the suffixes IE and SE in Table 6. The suffix SE refers to a shared embedding layer that learns to encode both the condition and the input sample, while IE indicates the use of separate embedding layers for each sample. One of the main advantages of an end-to-end embedding layer is the ability to learn distinct embeddings for different input samples, which can be achieved with two individual embedding layers (Devlin et al., 2018). In this configuration, the network

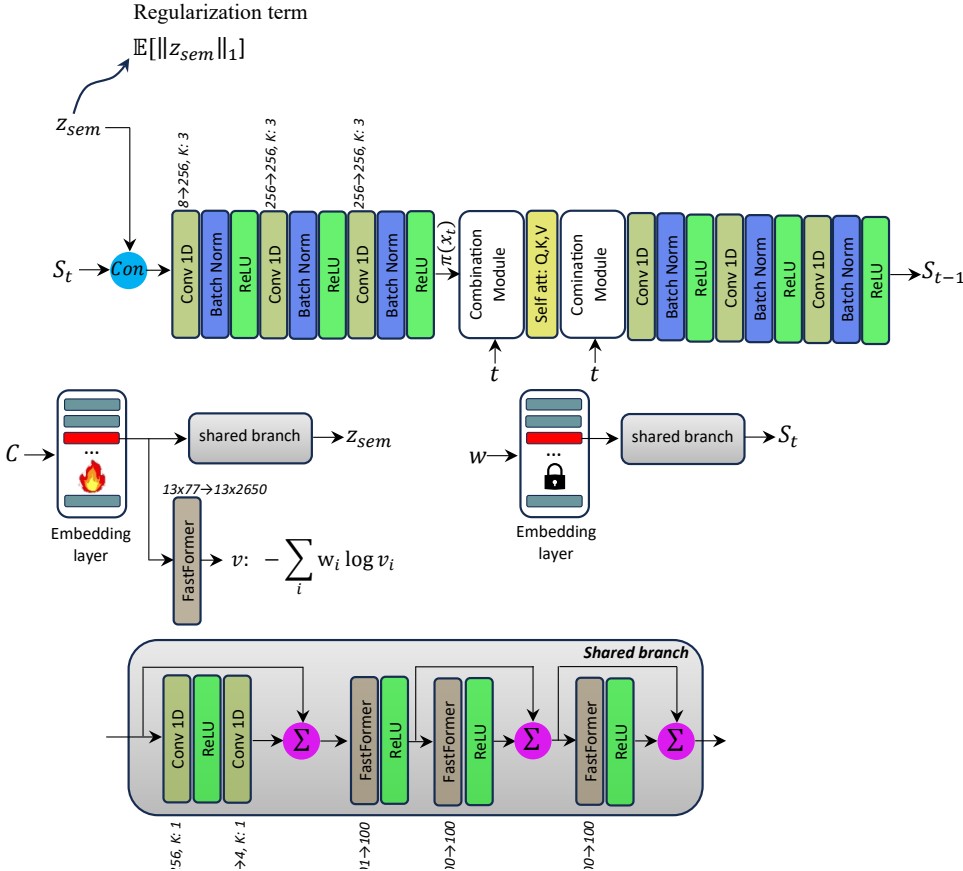

Figure 5: Implementation details for extending the strategy proposed in ReCondSE (Mittal et al., 2023) for categorical data. The embedding layer is supervised with the strategy proposed in (Lin et al., 2023;?), we do not consider the cycle embedding for the entail of this model.

| | Our method | | RCondSE (Mittal et al., 2023) | | RCondIE (Mittal et al., 2023) | | DiffAESE (Bahri et al., 2021) | | DiffAEIE (Devlin et al., 2018) | | InfoDiffusionSE (Devlin et al., 2018) | | InfoDiffusionIE (Chen et al., 2024b) | |
|---|---|---|---|---|---|---|---|---|---|---|---|---|---|---|
| **Heart failure prediction** | | | | | | | | | | | | | | |
| | AUROC | AUCPR | AUROC | AUCPR | AUROC | AUCPR | AUROC | AUCPR | AUROC | AUCPR | AUROC | AUCPR | AUROC | AUCPR |
| Support Vector | 93.80±0.43 | 19.14±5% | 91.56±0.76 | 14.78±6% | 89.75±0.49 | 13.34±5% | 90.42±0.85 | 14.41±7% | 90.62±0.44 | 14.93±4% | 90.10±0.52 | 14.51±8% | 91.73±0.84 | 15.82±9% |
| Gaussian Process | 92.96±0.66 | 17.93±5% | 91.16±0.46 | 16.33±5% | 91.97±0.35 | 17.57±4% | 89.48±0.37 | 13.86±4% | 91.17±0.74 | 15.07±6% | 89.04±63 | 13.18±5% | 91.07±0.63 | 15.81±6% |
| Random Forest | 90.26±0.54 | 13.24±4% | 88.31±0.66 | 11.71±5% | 89.41±0.54 | 13.06±6% | 90.85±0.63 | 14.98±6% | 91.47±0.75 | 16.84±7% | 91.70±0.85 | 16.41±7% | 90.64±0.72 | 14.44±6% |
| Ada Boost | 88.76±0.71 | 11.96±6% | 87.11±0.64 | 10.59±6% | 86.21±0.54 | 09.66±5% | 86.46±0.57 | 09.97±4% | 85.09±0.38 | 08.96±5% | 87.41±0.62 | 17.72±5% | 86.10±0.41 | 09.39±6% |
| GaussianNB | 90.32±0.57 | 14.17±6% | 88.21±0.74 | 11.26±6% | 87.93±0.36 | 11.34±4% | 87.45±0.44 | 11.221±4% | 87.37±0.54 | 10.59±5% | 85.68±0.66 | 09.62±7% | 88.49±0.82 | 11.60±11% |
| | Av.Kappa | Av.BF1 | Av.BF1 | Av.BF1 | Av.Kappa | Av.BF1 | Av.Kappa | Av.BF1 | Av.Kappa | Av.BF1 | Av.Kappa | Av.BF1 | Av.Kappa | Av.BF1 |
| | 0.882±9% | 0.897 | 0.862±9% | 0.873±9% | 0.831±9% | 0.849±9% | 0.756±9% | 0.784±9% | 0.857±9% | 0.741 | 0.793 | 0.802±9% | 0.896±9% | 0.773 |
| **Lung cancer prediction** | | | | | | | | | | | | | | |
| Support Vector | 93.73±0.46 | 0.81±6% | 90.49±0.53 | 0.53±4% | 89.70±0.52 | 0.50±6% | 90.07±0.77 | 0.51±6% | 89.32±0.49 | 0.48±4% | 89.68±0.63 | 0.49±5% | 88.82±0.64 | 0.46±6% |
| Gaussian Process | 90.94±0.69 | 0.56±5% | 87.57±0.44 | 0.41±5% | 87.81±0.47 | 0.42±4% | 87.84±0.51 | 0.42±6% | 88.27±0.37 | 0.43±4% | 88.31±0.78 | 0.44±6% | 87.67±0.66 | 0.41±5% |
| Random Forest | 92.85±0.34 | 0.71±5% | 88.35±0.65 | 0.44±6% | 88.78±0.40 | 0.45±4% | 89.62±0.59 | 0.49±6% | 90.20±0.43 | 0.52±5% | 89.04±0.78 | 0.47±6% | 89.38±0.62 | 0.48±6% |
| Ada Boost | 90.35±0.67 | 0.53±5% | 85.65±0.69 | 0.36±11% | 85.91±0.52 | 0.36±6% | 87.05±0.38 | 0.40±4% | 87.43±0.39 | 0.41±5% | 88.14±0.64 | 0.43±4% | 88.46±0.62 | 0.44±6% |
| GaussianNB | 90.10±0.50 | 0.51±6% | 88.15±0.47 | 0.43±4% | 88.43±0.43 | 0.44±5% | 85.74±0.78 | 0.36±6% | 86.52±0.66 | 0.38±6% | 87.52±0.61 | 0.41±6% | 88.25±0.54 | 0.43±5% |
| | Av.Kappa | Av.BF1 | Av.BF1 | Av.BF1 | Av.Kappa | Av.BF1 | Av.Kappa | Av.BF1 | Av.Kappa | Av.BF1 | Av.Kappa | Av.BF1 | Av.Kappa | Av.BF1 |
| | 1.40±7% | 47.44±0.68 | 1.01±8% | 46.03±0.82 | 0.74±11% | 44.42±9% | 0.79±6% | 44.82±0.74 | 0.72±7% | 44.25±0.42 | 0.51±6% | 0.802±9% | 0.68±11% | 43.92±0.71 |

Table 6: Performance comparison of our method with diffusion-based representation learning methods

can automatically learn the same embedding for samples that exhibit a high correlation. Our goal is to assess whether this type of embedding can improve the performance of our competing strategies, which already benefit from end-to-end learning of the embedding layers.

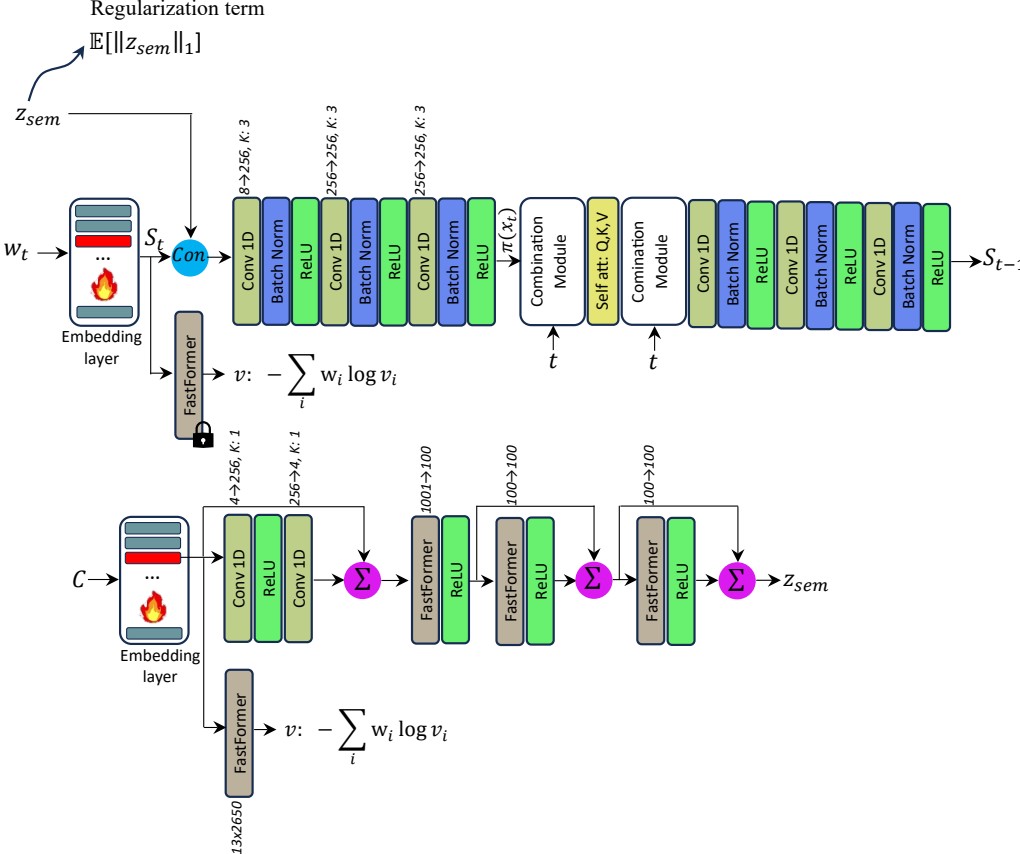

Figure 6: Implementation details for extending the strategy proposed in ReCondIE (Mittal et al., 2023) for categorical data

As shown in Table 6, our method achieves improvements of 2.24%, 1.83%, 2.95%, 2.33%, 2.10%, and 2.07% in AUROC compared to the best results of RCondSE, RCondIE, DiffAESE, DiffAEIE, InfoDiffusionSE, and InfoDiffusionIE for the heart failure prediction task. For lung cancer prediction, our method outperforms the best results of the competing algorithms by 3.24%, 4.03%, 3.66%, 3.53%, 4.05%, and 4.35%. The improvement in lung cancer prediction is notably greater than that in the heart failure prediction, likely because the number of patients with lung cancer in our database is significantly smaller than that for heart failure. Consequently, the benefits of the probabilistic representation enabled by cycle conditioning are more pronounced in lung cancer prediction.

In contrast, the competing strategies exhibited similar performance, which may be attributed to our use of the same configuration for most of their components. This suggests that the regularization techniques introduced in (Mittal et al., 2023; Preechakul et al., 2022; Wang et al., 2023b) have a limited impact on the representations learned by the conditioning encoder. It is possible that different architectures would enhance their effectiveness when applied in their original form, which is specifically designed for image-related tasks. Additionally, their lower performance on categorical data may stem from the lack of a specialized design to extract versatile information from the conditioning encoder.

## C  Cycle consistency vs disentangled guidance

We propose our cycle conditioning strategy as a method to enhance the representations learned by diffusion models. Additionally, we suggested that this cycle conditioning can prevent the diffusion process from

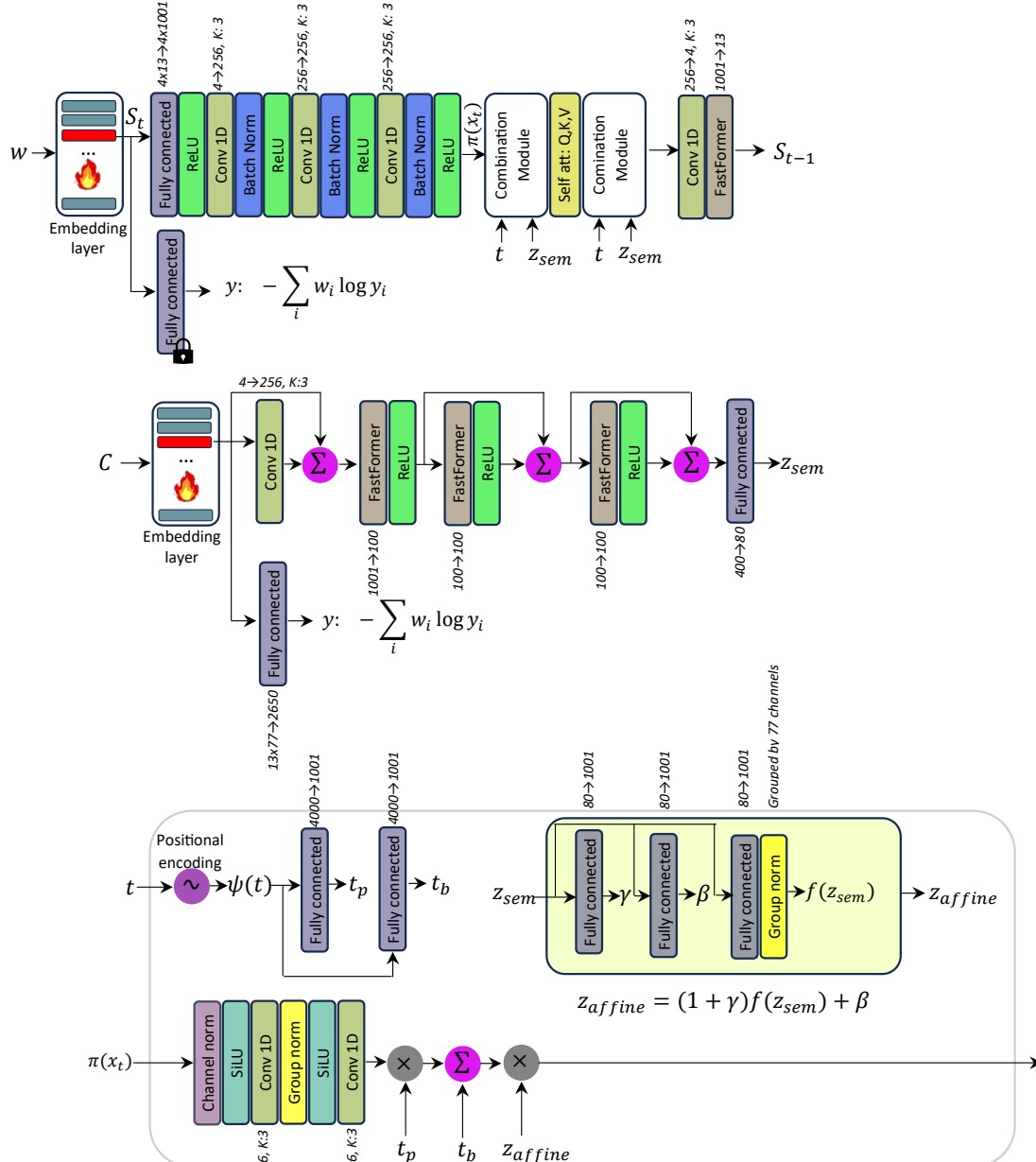

Figure 7: Implementation details for extending the strategy proposed in DiffAEIE (Preechakul et al., 2022) for categorical data. The embedding layer is supervised with the strategy proposed in (Lin et al., 2023)

overlooking the condition when the hidden condition is supervised by extra spelling information of the original time series. Another strategy to address this issue is disentangled guidance, adapted from the work of (Bhunia et al., 2023). In this approach, the condition is preserved by replacing $S_t$ with an all-zero tensor and supervising the network to reconstruct the condition from this all-zero tensor combined with the latent condition. This reconstruction is facilitated by the combination modules in the denoising network and is used in conjunction with the conventional loss functions of diffusion models. The strategy and the definition of the combination module are illustrated in Figure 4.

Tables 7 and 8 compare the performance of our method with the alternative strategy of disentangled guidance. As shown, using a supervised cycle conditioned network, we achieved 3.53% and 3.00% higher AUROC

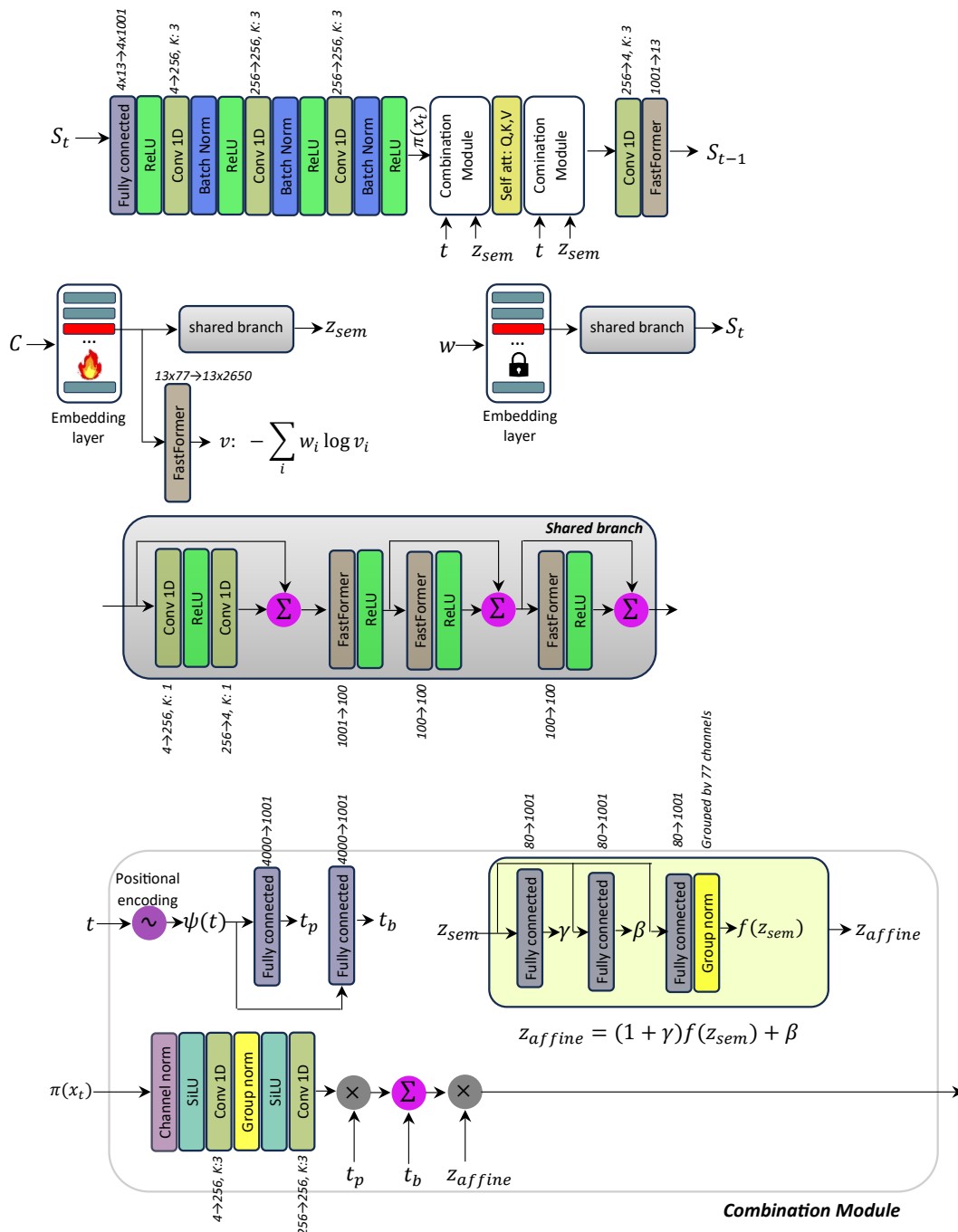

Figure 8: Implementation details for extending the strategy proposed in DiffAESE (Preechakul et al., 2022) for categorical data. The embedding layer is supervised with the strategy proposed in (Lin et al., 2023), we do not consider the cycle embedding for the entail of this model.

compared to the best results obtained with disentangled guidance on the two downstream tasks, respectively. Furthermore, the mean AUROC of our method across different downstream classifiers is 2.96% higher than that of the disentangled conditioning.

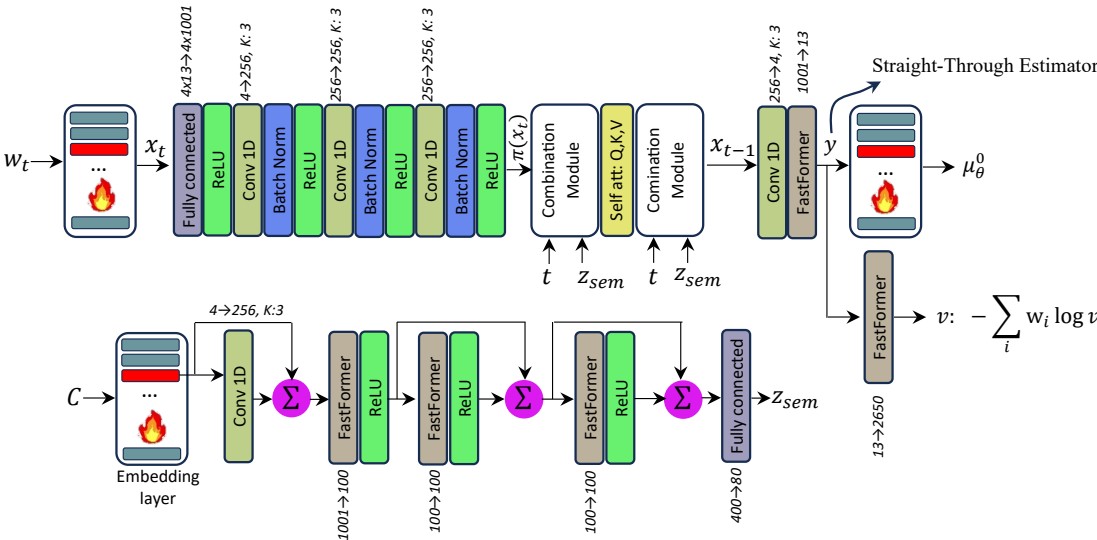

Figure 9: Cycle embedding of diffusion model, according to our experiments this structure has significant difficulty with convergence.

This improvement may be attributed to the fact that, in disentangled guidance, the condition is processed differently when provided without the noisy sample compared to when it is combined with the noisy sample. In contrast, our method employs a supervision approach where both the input and the denoised sample follow the same pathway within the network. This design helps avoid conflicts between the learned representations in the input of the denoising model and those maintained in its output.

Additionally, disentangled conditioning does not allow for generating multiple representations from a single conditioning sample. Our method addresses this limitation by producing diverse representations from a single sample. This capability is especially advantageous when there are not enough labeled samples to adequately train a downstream classifier.

Moreover, disentangled conditioning introduces extra supervision on an intermediate layer. This supervision involves estimating the elements of the spelling vector from an intermediate layer following the combination modules. Such supervision can lead to sparsity in that intermediate layer. Since this layer is located close to the model's output, it may negatively impact the performance of the denoising model in estimating the noise in its input sample.

|  | Cycle conditioning | | Disentangled guidance | |
|---|---|---|---|---|
|  | AUROC | AUCPR | AUROC | AUCPR |
| Support Vector | 91.81±0.61 | 12.49±5% | 87.57±0.52 | 09.23±6% |
| Gaussian Process | 90.96±0.62 | 13.70±5% | 86.14±0.72 | 08.16±8% |
| Random Forest | 89.48±0.37 | 14.21±5% | 88.28±0.69 | 13.61±5% |
| Ada Boost | 85.12±0.73 | 08.62±6% | 84.01±0.43 | 08.16±4% |
| GaussianNB | 89.20±0.76 | 08.41±7% | 85.73±0.82 | 08.19±7% |

Table 7: Comparing cycle conditioning and disentangled guidance for heart failure prediction

|  | Cycle conditioning | | Disentangled guidance | |
|---|---|---|---|---|
|  | AUROC | AUCPR | AUROC | AUCPR |
| Support Vector | 90.51±0.64 | 0.54±9% | 86.19±0.55 | 0.41±6% |
| Gaussian Process | 88.79±0.86 | 0.56±8% | 87.86±0.63 | 0.52±4% |
| Random Forest | 91.35±0.48 | 0.59±5% | 88.35±0.77 | 0.48±4% |
| Ada Boost | 86.60±0.57 | 0.49±6% | 86.92±0.85 | 0.51±6% |
| GaussianNB | 84.39±0.77 | 0.43±6% | 85.62±0.61 | 0.46±5% |

Table 8: Comparing cycle conditioning and disentangled guidance for lung cancer prediction

## D   Class-inspired representation learning meets the cycle conditioning

In this section, we propose an end-to-end class-inspired representation learning method that utilizes class labels from a subset of samples to develop robust representations for all samples in the database. This approach is particularly valuable for applications where the number of training samples with specific labels is

very limited. Unlike multi-task representation learning, the representations learned by this model focus on the characteristics most relevant to predicting a particular label.

While this method can eliminate the need for a downstream classifier by integrating the classifier into the representation learning framework, we found that training an additional classifier on the class-specific representations generated by this model is significantly more effective than relying on the model's internal classifier.

For this modeling, we first consider a lower bound estimation of our previously proposed method for a batch of data that includes at least one sample diagnosed with a specific disease of interest. This can be any kind of disease including heart failure or lung cancer that we already considered as downstream tasks of our model. In this case, the variational bound of our model can be written as follows:

$$
\begin{aligned}
\log p_\theta(S, y) &= \mathbb{E}_q \left[ -\log \frac{p_\theta(S_{0:T}, y)}{q(S_{1:T}|S_0, y)} \right] \\
&= \mathbb{E}_q \left[ -\log p_\theta(S_T, y) - \sum_{t \geq 1} \log \frac{p_\theta(S_{t-1}|S_T, y)}{q(S_t|S_{t-1}, y)} \right] \\
&= \mathbb{E}_q \left[ -\log p_\theta(S_T, y) \right] + \sum_{t \geq 1} \mathbb{E}_{q(S_t|S_{t-1}, y)} \left[ -\log \frac{p_\theta(S_{t-1}|S_T, y)}{q(S_t|S_{t-1}, y)} \right] \\
&= \mathbb{E}_q \left[ -\log p(S_T) - \log p_\theta(y) \right] \\
&\qquad\qquad + \sum_{t \geq 1} \mathbb{E}_{q(S_t|S_{t-1}, y)} \left[ -\log p_\theta(S_{t-1}|S_t, y) + \log q(S_t|S_{t-1}, y) \right] \\
&= \sum_{t \geq 1} \mathbb{E}_{q(S_t|S_{t-1}, y)} \left[ -\frac{1}{T} \log p(S_T) - \frac{1}{T} \log p_\theta(y) - \log p_\theta(S_{t-1}|S_t, y) + \log q(S_t|S_{t-1}, y) \right]
\end{aligned}
\tag{16}
$$

$p_\theta(S_T, y) = p(S_T)p_\theta(y)$ as $S_T$ and $y$ are independent and the distribution of $S_T$ is already known and therefore is not dependent on $\theta$.
This equation measures an accumulated reconstruction loss over time steps of the denoising process and therefore can be approximated as follows:

$$
\begin{aligned}
\log p_\theta(S, y) &\approx \sum_{t \geq 1} D_{KL}(q(S_{t-1}|S_t, y, S_0) \| p_\theta(S_{t-1}|S_t, y)))) \\
&= \sum_{t \geq 1} \mathcal{L}_t(S, y)
\end{aligned}
\tag{17}
$$

For data batches where none of the training samples are diagnosed with the intended disease that we aim for the representation to learn, the information from the labels is treated as a latent variable. This approach allows us to estimate the lower bound for these samples using the following formula:

$$
\begin{aligned}
\log p_\theta(S) &= \sum_{t \geq 1} \mathbb{E}_{q(y, S_t|S_{t-1})} \left[ -\frac{1}{T} \log p(S_T) - \frac{1}{T} \log p_\theta(y) - \log p_\theta(S_{t-1}|S_t, y) + \log q(y, S_t|S_{t-1}) \right] \\
&= \sum_{t \geq 1} \mathbb{E}_{q(y|S_{t-1})} \left[ \mathbb{E}_{q(S_t|y, S_{t-1})} \left[ -\frac{1}{T} \log p(S_T) - \frac{1}{T} \log p_\theta(y) \right. \right. \\
&\qquad\qquad\qquad\qquad\qquad\qquad \left. \left. - \log p_\theta(S_{t-1}|S_t, y) + \log q(y, S_t|S_{t-1}) \right] \right]
\end{aligned}
\tag{18}
$$

Using the chain rule, we have $q(y, S_t|S_{t-1}) = q(y|S_{t-1})q(S_t|y, S_{t-1})$. Therefore, this equation can be reformulated as follows:

$$
\begin{aligned}
\log p_\theta(S) &= \sum_{t \geq 1} \mathbb{E}_{q(y|S_{t-1})}\Bigg[\mathbb{E}_{q(S_t|y,S_{t-1})}\Big[ -\frac{1}{T}\log p(S_T) - \frac{1}{T}\log p_\theta(y) \\
&\qquad\qquad\qquad\qquad\qquad - \log p_\theta(S_{t-1}|S_t, y) + \log\big(q(y, S_t)q(S_t|S_{t-1}, y)\big)\Big]\Bigg] \\
&= \sum_{t \geq 1} \mathbb{E}_{q(y|S_{t-1})}\Bigg[\mathbb{E}_{q(S_t|y,S_{t-1})}\Big[ -\frac{1}{T}\log p(S_T) - \frac{1}{T}\log p_\theta(y) \\
&\qquad\qquad\qquad\qquad\qquad - \log p_\theta(S_{t-1}|S_t, y) + \log q(y|S_{t-1}) + \log q(S_t|S_{t-1}, y)\Big]\Bigg]
\end{aligned}
\tag{19}
$$

Therefore, the equation can be reformulated as:

$$
\begin{aligned}
\log p_\theta(S) &= \sum_{t \geq 1}\sum_y q(y|S_{t-1})\Big( -\mathbb{E}_{q(S_t|y,S_{t-1})}\Big[ -\log p(S_T) - \log p_\theta(y) \\
&\qquad\qquad\qquad\qquad - \log p_\theta(S_{t-1}|S_t, y) + \log q(y|S_{t-1}) + \log q(S_t|S_{t-1}, y)\Big]\Big) \\
&= \sum_{t \geq 1}\sum_y q(y|S_{t-1})\Big( -\mathcal{L}_t(S, y) + \mathbb{E}_{q(S_t|y,S_{t-1})}\big[q(y|S_{t-1})\big]\Big)
\end{aligned}
\tag{20}
$$

Where $\sum_y q(y|S_{t-1})\Big(\mathbb{E}_{q(S_t|y,S_{t-1})}\big[q(y|S_{t-1})\big]\Big)$ is the entropy of $q(y|S_{t-1})$ and therefore measures how spread out the probabilities are in this specific distribution.

From this equation, the distribution $q(y|S_{t-1})$ can formulated as a discriminative classifier, however similar to our cycle conditioned strategy it has to implemented in a cycle conditioned manner as it depends to the output of the denoising model. Therefore, it is beneficial to reimplement both the cycle and the label conditioning elements with a simple shared common encoder with different heads. In practice we estimate the label $y$ by applying two fully connected layers after $E_\theta(.)$, with 30 unites per each layer.

Our experiments show that the class-inspired cycle conditioning achieves an AUROC of 94.18% on the heart failure prediction task when using Support Vector Machines as the downstream classifier. This represents an improvement of 1.01% over the maximum AUROC value obtained with the vanilla version of our cycle conditioning method. Additionally, a similar experiment was conducted to predict heart failure within a four-month time window, using the same data split as described in the main paper.

## E  Exploring Data Size Variance

In this section, we evaluate how the size of the training data impacts the expressiveness of the representations learned by our diffusion model. We consider heart failure prediction as the downstream task for this experiment. To this end, we utilize two distinct training sets: one from the MIMIC IV database and another from the MIMIC III database, both of which are used to train our representation learning model. After training, we employ these pre-trained models to project both the training and test samples into an embedding space. The test set is selected from the MIMIC IV database, and the data split follows the same procedure outlined in the main paper.

Our training sets from MIMIC-III exclusively contain ICD-9 codes, while our test samples from MIMIC-IV comprise both ICD-9 and ICD-10 codes. To align with the vocabulary of our MIMIC-IV database, we index

the tokens from MIMIC-III based on their first three characters, matching them with the corresponding tokens in the vocabulary of the MIMIC-IV database.

We compared the performance of our method with SwitchTab (Wu et al., 2024) using the two databases. The results, presented in Table 9, demonstrate that our method outperforms this competing strategy, even when the number of training samples is significantly reduced. Notably, when trained on only 5% of the training data, our method achieved an accuracy comparable to many competing strategies that were trained on the entire training set of the MIMIC IV database (see Table 7).

| | MIMIC IV | | MIMIC III | |
|---|---|---|---|---|
| | AUROC | AUCPR | AUROC | AUCPR |
| Our method | $91.81\pm0.61$ | $12.49\pm5\%$ | $81.72\pm0.52$ | $05.23\pm6\%$ |
| SwichTab | $90.39\pm0.76$ | $09.57\pm6\%$ | $65.88\pm0.96$ | $02.93\pm11\%$ |

Table 9: Performance comparison of our method on two databases with varying sizes: We employ MIMIC III and MIMIC IV for training our diffusion-based representation learning method.

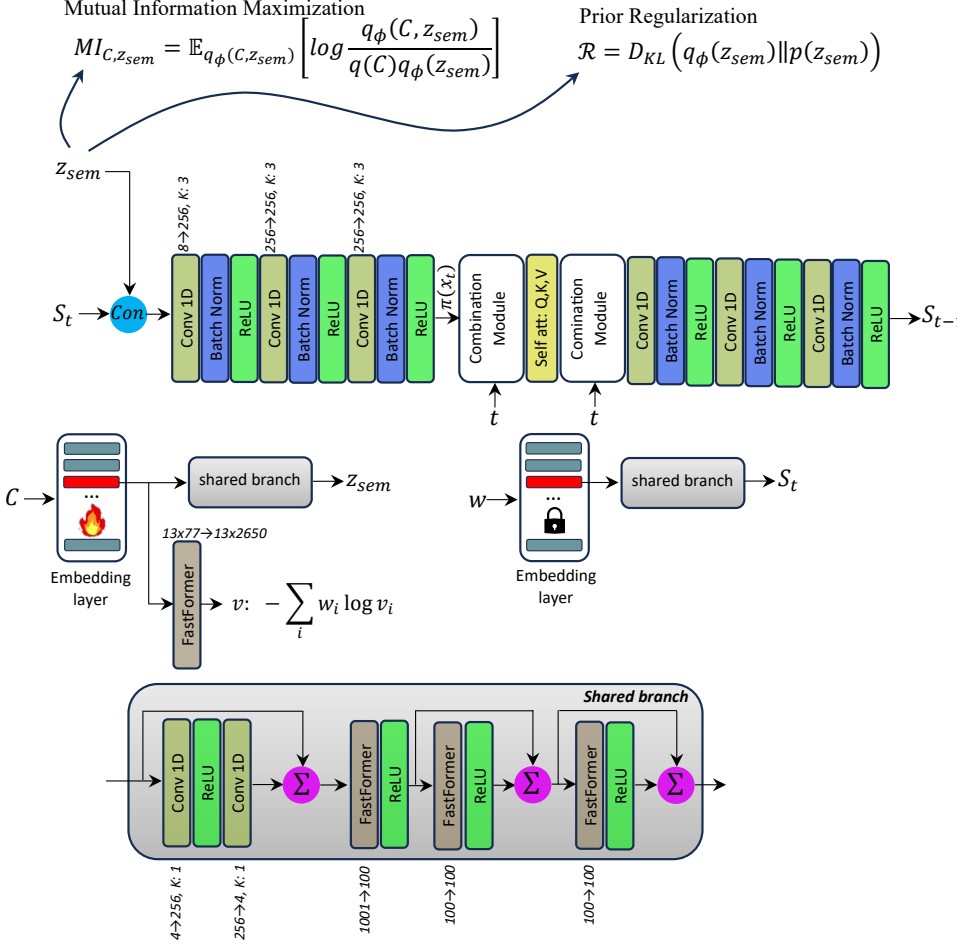

Figure 10: Implementation details for extending the strategy proposed in InfoDiffusionSE (Wang et al., 2023b) for categorical data.

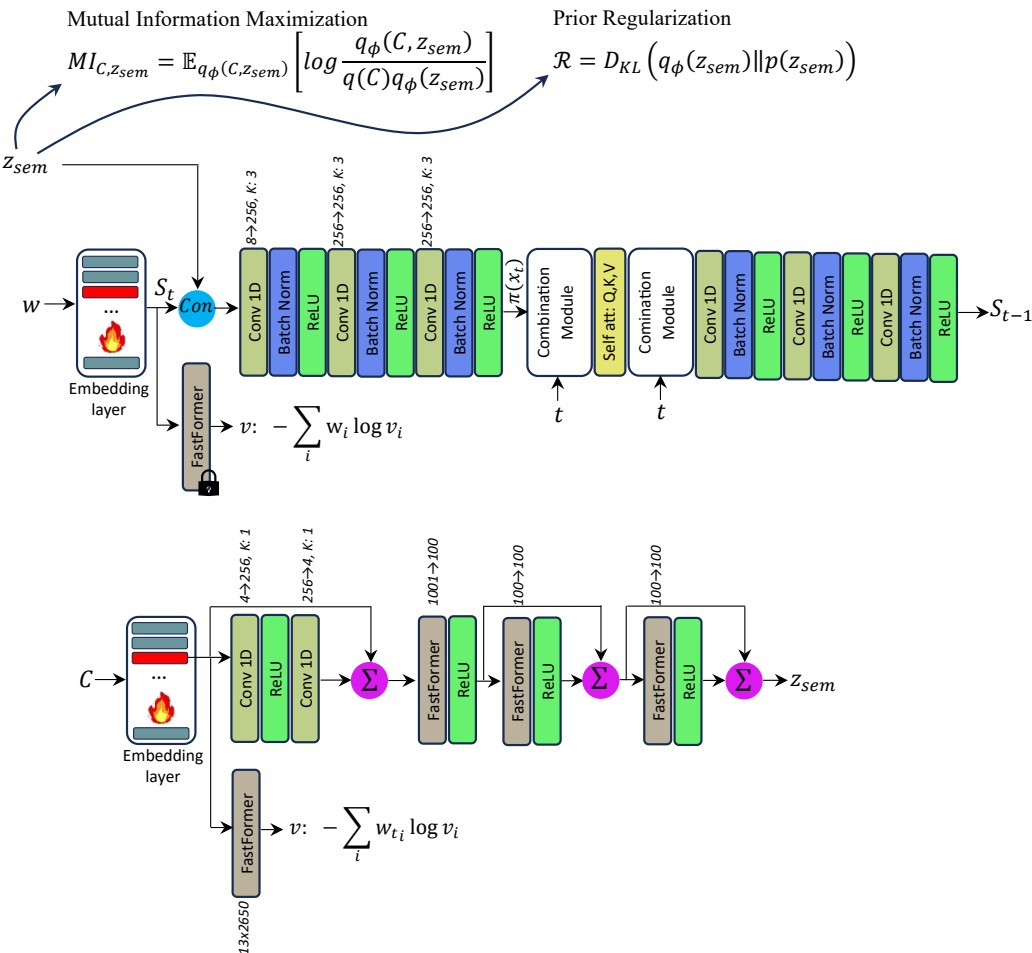

Figure 11: Implementation details for extending the strategy proposed in InfoDiffusionIE (Wang et al., 2023b) for categorical data.

## F  Notations

In Table 4, Em-MLM denotes continuous data from the embedding layer trained using mask modeling, OutBERT-MLM represents continuous data from one layer before the output of the BERT model trained using mask modeling, and EM-ConAu signifies continuous data from the embedding layer of the BERT model trained using the method proposed in (Chen et al., 2024b).

## G  Our data transformation strategy versus BERT embedding layer for downstream tasks

In this section, we compare the performance of our proposed data transformation module with that of the BERT embedding layer, which was trained using masked language modeling, on two downstream tasks: heart failure prediction and lung cancer prediction[4]. As shown in Tables 10 and 11, our method consistently outperforms the BERT embedding layer across different classifiers. However, the extent of this improvement varies by task, with the performance gain being smaller for lung cancer prediction compared to hear failure

---

[4]Detailed results for the BERT embedding layer, along with comparisons to other classification models, are presented in Table 1.

①*Unique characters in the vocabulary*
$A$, $B$, ...      , $Z$, 0, 1, ... ,9

②*Assigning an integer to each character*
$Z$, $Y$, ...      , $A$, 9, 8, ... ,0
↓   ↓            ↓   ↓   ↓        ↓
35, 34,  ...    , 10, 9,  8,  ... ,0

③ *Representing each token by*
   *concatenating its integers*

$$\begin{bmatrix} 'H73' & 'X45' & \cdots \\ \vdots & \vdots & \ddots \\ 'A08' & 'Y55' & \cdots \end{bmatrix} \quad \begin{matrix} 'H73' & \rightarrow 1773 \\ 'X45' & \rightarrow 3345 \\ \vdots & \vdots \end{matrix}$$

④*Creating multi-hot Spelling vector*

$$u = [0 \quad 0 \quad \cdots \quad 1 \overset{\cdots}{\leftarrow} \quad 0] \quad _{1773\text{-th entry}}$$

Figure 12: The vector used to supervise the latent code of the conditioning sample is derived from the spelling of the tokens in w. This spelling information is represented by the methodology illustrated in this figure. The initial direction towards each spelling index (e.g., 1773) is learned by an element of the dictionary ($d_x$), where $x$ represents the index.

prediction. We attribute this difference to the fact that masked language modeling does not capture time-specific nuances in the embedding layer, which may limit its performance, especially in tasks that rely more on time-specific features. When comparing the performance of our embedding layer with that of our model without spelling supervision, we observe that the performance of the diffusion model with cycle conditioning alone is even worse than that of the input to our diffusion model. This is because cycle conditioning without external supervision causes the diffusion model to forget much of the conditioned information, as many of these details change in each iteration of training.

|  | BERT embedding layer | | Our DT strategy | |
|---|---|---|---|---|
|  | AUROC | AUCPR | AUROC | AUCPR |
| Support Vector | 66.49±0.44 | 03.15±4% | 76.21±0.36 | 04.29±6% |
| Gaussian Process | 64.10±0.45 | 02.86±4% | 77.33±0.66 | 06.37±5% |
| Random Forest | 67.63±0.44 | 03.31±4% | 78.04±0.63 | 05.28±6% |
| Ada Boost | 64.28±0.47 | 02.88±6% | 74.34±0.51 | 06.72±5% |
| GaussianNB | 60.58±0.68 | 02.54±6% | 70.04±0.64 | 06.14±5% |

Table 10: Our data transformation strategy versus BERT embedding layer for heart failure prediction. DT refers to Data Transformation

|  | BERT embedding layer | | Our DT strategy | |
|---|---|---|---|---|
|  | AUROC | AUCPR | AUROC | AUCPR |
| Support Vector | 83.47±0.77 | 0.34±9% | 84.13±0.66 | 0.38±5% |
| Gaussian Process | 80.14±0.37 | 0.31±4% | 85.47±0.81 | 0.27±7% |
| Random Forest | 84.19±0.61 | 0.38±6% | 85.73±0.66 | 0.32±7% |
| Ada Boost | 81.25±0.43 | 0.24±4% | 82.28±0.72 | 0.37±5% |
| GaussianNB | 82.12±0.38 | 0.29±4% | 83.71±0.33 | 0.30±6% |

Table 11: Our data transformation strategy versus BERT embedding layer for lung cancer prediction. DT refers to Data Transformation.

## H  Computational Complexity

In this section, we provide an analysis of the computational complexity of our method compared to its competing strategies. Our method consists of two primary components. The first part is a data transformation strategy that takes an input sample in the form of a one-hot vector of size $1 \times \mathcal{W}$, where $\mathcal{W}$ represents the total number of unique tokens in the vocabulary, and transforms it into a continuous hidden representation of size $1 \times m$, where $m$ is the size of the embedding layer. The core of this transformation is a matrix of size $\mathcal{W} \times m$. Therefore, the total number of scalar operations required for multiplying this matrix with the input sample is $O(\mathcal{W} \times m)$. For a time series with $r$ time steps, the overall complexity becomes $O(r \times \mathcal{W} \times m)$. In contrast, the BERT embedding layer is a lookup table with learnable parameters, where the lookup operation takes $O(1)$ time to retrieve the learned embedding for each token in the input sequence. For a time series of $r$ individual time steps, the computational complexity of BERT's embedding is $O(r \times q)$, where $q$ is the number of tokens in each of the $r$ time steps. Since $\mathcal{W} \gg q$, the computational cost of our method is higher than that of the BERT embedding layer, and this cost scales with the size of the vocabulary. However, this comparison assumes that the concatenation of feature embeddings for the entire sequence of tokens in BERT is not prohibitively large and can be directly used as the embedding representation. In cases where the size of

the embedding layer for each token needs to be large enough to capture the necessary contextual information, we must introduce an additional linear layer to project the embeddings into a lower-dimensional space. This transformation involves multiplying the embeddings with a weight matrix, which results in an additional computational cost of $O(r \times q \times m_H \times m)$, where $m_H$ is the dimension of each embedding before the linear projection, and $m$ is the size of the output embedding space after projection.

The second part of our method is a conditional diffusion model, where the representation learned from the conditioning sample is utilized as the final representation of the input sample. Therefore, the computational cost of this part depends heavily on the computational cost of the conditioning encoder. Central to our conditioning encoder is a FastFormer module (Wu et al., 2021), which leverages the additive attention mechanism to reduce the quadratic complexity of Transformers to linear complexity. In Transformers, a sequence of length $m$ requires $m \times m$ pairwise computations to account for all combinations of queries, keys, and values. Hence, it has $O(m^2)$ computational complexity. In contrast, FastFormer avoids the pairwise computation by utilizing additive operations to aggregate attention. As a result, the computational complexity required for computing attention is reduced to $O(m)$. This represents a significant advantage in the computational efficiency of our method compared to standard Transformer-based models like SwitchTab 2024, especially when the size of the embedding layer scales up.

