# OpenReview forum: "Cycle Conditioning for Robust Representation Learning from Categorical Data"
_TMLR — Accepted by TMLR_

### Review · Reviewer_FezA · 2024-12-23

**Summary Of Contributions:**

In this work, the authors present a diffusion-based representation learning framework for categorical (discrete) time series data. Their approach incorporates two key components: (1) a data transformation module that maps the multi-hot representations of tokens at each time step into continuous vectors, while preserving token set similarities, and (2) a diffusion-based representation learning component designed to reconstruct the original sequence and extract robust features as conditioning embeddings. To enhance representation learning, the framework introduces three regularization techniques—cycle conditioning to ensure consistency in extracted features, spelling supervision to maintain conditioning influence during denoising, and latent space navigation to address sparsity in spelling information by introducing learnable directional codes.

Such a framework significantly improves over SoTA methods, particularly in predictive healthcare tasks on MIMIC-III and MIMIC-IV datasets.​

**Audience:**

Yes

**Broader Impact Concerns:**

No ethical concerns

**Claims And Evidence:**

Yes

**Requested Changes:**

1) More ablation studies:
(1) An ablation study on the data transformation module to compare the performance of the proposed embeddings against MLM-based embeddings for downstream tasks. (2) A comparison between cycle conditioning and explicit alignment of $E_\theta(S_0)$ and $E_\theta(\hat{S}_0(S_t))$ (e.g. MSE loss between clean & noisy representations) to assess the necessity of cycle conditioning.

2) Writing improvements:
(1) Include a brief preliminary explanation of time series with categorical data to help readers understand the context better. Clarify the distinction between "time step" as used in time series analysis and DMs. (2) Provide relevant citations for the second part of the method to better contextualize its novelty and highlight connections to existing work. Without this, the method may appear overly ad-hoc and harder to follow.

**Strengths And Weaknesses:**

### Strengths:

1) The first component of the proposed framework, which transforms multi-hot representations of tokens into continuous embeddings, is well-motivated and clearly explained. Such design takes the gap between traditional text embeddings and categorical time series data into consideration, while also maintaining robustness against small token modifications. The effectiveness is validated in Section 6, where the authors' method shows superior Spearman correlation compared to embeddings from Masked Language Modeling (MLM), highlighting its ability to capture contextual relationships.

2) The authors incorporate practical strategies to optimize representation learning. For example, cycle conditioning ensures consistency in extracted representations between clean and noisy samples, while spelling supervision guarantees that the learned embeddings retain critical spelling information, thereby preventing the diffusion model(DM) from neglecting the conditioning representation. Furthermore, the authors conduct comprehensive comparative studies against SoTA methods, consistently demonstrating significant improvements in predictive tasks for healthcare applications​.

### Weaknesses:

1) My primary concern lies in the design choice of learning the representations $z_\text{sem}$ solely as optimal conditioning for the DM, rather than aligning them with its intermediate features. This is because conditioning accounts for a relatively small aspect of the DM, while intermediate features appear to be more critical for reconstruction. While the authors argue that *"diffusion models do not include a bottleneck layer, so attention has shifted to conditional diffusion models"*, this is inconsistent with existing UNet-based DMs, which possess a bottleneck layer that could serve as a robust representation space (e.g. https://arxiv.org/abs/2112.03126, https://arxiv.org/abs/2303.09769). Similarly, features from the middle blocks of transformer-based DMs are highly informative, even without an explicit bottleneck (e.g. https://arxiv.org/abs/2401.14404, https://arxiv.org/abs/2410.06940). Though Table 5 shows the superiority of the condition-based method towards those that use unconditional DMs, its effectiveness in more general domains, such as vision or language tasks, remains unverified.

2) Due to the training objective discussed above, the authors need to introduce additional ad-hoc mechanisms, such as cycle conditioning to ensure robustness to noise (a property naturally achieved by intermediate features) and spelling supervision with latent navigation to sufficiently train the conditioning representations. Other approaches may not require such modifications. Also, the ablation study in Table 9 indicates that cycle conditioning contributes much less than spelling supervision, raising questions about the necessity and effectiveness of such a design.

3) The iterative construction of the mapping in the data transformation step, while intuitive, introduces potential computational complexity, as it involves inverting a $d\times d$ matrix at each epoch, where $d$ represents the total number of tokens.

---

> ### Author Response · Authors · 2025-01-01
>
> We thank the reviewer for the useful feedback. We address your comments below.
>
> While the authors argue that "diffusion models do not include a bottleneck layer,…
>
> In practice, representation learning aims to create a compact representation of input samples that retains critical information about the original data while allowing downstream tasks to be lightweight and efficient. Compactness is crucial because it ensures that the learned representations can be processed effectively by simpler classifiers, without requiring extensive computational overhead or deep learning-based models that are already well-proven to be effective to do the same task on the original samples. Therefore if the task is to extract the representation and supervise it for another task, this representations should be compact and more efficients.
> The notion of a "bottleneck" in generative models such as GANs and VAEs is typically associated with a highly compact intermediate layer. For instance, in StyleGAN, this bottleneck has a size of 512, while for VAEs (as noted in https://arxiv.org/abs/2112.10752), the size can be as low as 4,096 for an input sample of size $512 \times 512$.
> However, in the context of U-Nets used in diffusion models, such a bottleneck is absent in the traditional sense. While the U-Net does reduce the spatial resolution of input samples (e.g., via pooling or strided convolutions), this reduction is much compensated by an increase in channel size. For instance, in DDPM (https://arxiv.org/abs/2006.11239), the intermediate feature maps for an input sample of size $256 \times 256$ have dimensions $16 \times 16 \times 256 = 65,536$, which is significantly larger than the compact bottlenecks found in GANs or VAEs. This dimensionality explains why works such as https://arxiv.org/abs/2112.03126 still require A deep neural network with substantial parameter counts to extract meaningful information from the intermediate representations. Our primary goal, in contrast, is to learn a \textit{low-dimensional meaningful representation} of the input sample that can be used effectively with lightweight classifiers (e.g., without the need for a deep, parametric model downstream), which is highly important for medical applications.
> It is worth to note that in case of Stable Diffusion, where the size of the U-Net’s input and intermediate layers is significantly smaller (e.g., $64 \times 64 \times 4$) than in traditional U-Nets used in diffusion models. However, it is important to clarify that this compact representation is a result of the VAE’s pretrained encoder, which projects the input image (e.g., of size $512 \times 512 \times 3$) into a compact latent space. This compression step, achieved by the VAE, is distinct from the U-Net’s architecture and the diffusion process itself. In this case, the learned representation’s compactness and structure are attributable to the VAE encoder’s bottleneck, rather than the U-Net employed by the diffusion model.
>
> To summarize, while intermediate features in U-Nets for diffusion models may contain useful information, they are not inherently compact nor optimized to serve as standalone, low-dimensional representations without further processing.
>
> We would also like to address your observation about the contribution of cycle conditioning appearing less significant compared to spelling supervision. While it is true that spelling supervision plays a dominant role, it is worth noting that even a modest improvement, such as the 2% gain achieved through cycle conditioning in our experiments, is highly impactful in domains like medical applications. For instance, when the task involves predicting the progression of a disease in a patient over the next few months, a marginal improvement in predictive accuracy can translate into meaningful real-world outcomes.
> Additionally, the comparison between using diffusion models with cycle conditioning but without spelling supervision, versus using spelling supervision alone, should not be taken to imply that cycle conditioning has a negligible effect. Our experiments demonstrate that without cycle conditioning, the network becomes overwhelmed by noisy samples, leading to less meaningful learned representations. In fact, incorporating cycle conditioning provided a 1.7% improvement in accuracy, reinforcing its value as part of the overall design.

---

> > ### Author Response · Authors · 2025-01-01
> >
> > the authors need to introduce additional ad-hoc mechanisms...
> >
> > We would also like to address your observation about the contribution of cycle conditioning appearing less significant compared to spelling supervision. While it is true that spelling supervision plays a dominant role, it is worth noting that even a modest improvement, such as the 2% gain achieved through cycle conditioning in our experiments, is highly impactful in domains like medical applications. For instance, when the task involves predicting the progression of a disease in a patient over the next few months, a marginal improvement in predictive accuracy can translate into meaningful real-world outcomes.
> >
> > Additionally, the comparison between using diffusion models with cycle conditioning but without spelling supervision, versus using spelling supervision alone, should not be taken to imply that cycle conditioning has a negligible effect. Our experiments demonstrate that without cycle conditioning, the network becomes overwhelmed by noisy samples, leading to less meaningful learned representations. In fact, incorporating cycle conditioning provided a 1.7% improvement in accuracy, reinforcing its value as part of the overall design.

---

> > > ### Author Response · Authors · 2025-01-01
> > >
> > > More ablation studies...
> > >
> > > This comparison has now been included in the revised manuscript. We show that our proposed embeddings consistently outperform the MLM-based embeddings across the evaluated tasks, demonstrating the effectiveness of our data transformation strategy. The updated results and analysis can be found in Page 26, Section G of the revised paper.
> > >
> > >
> > >  Comparison between cycle conditioning and explicit alignment of  ...
> > >
> > > Regarding your suggestion to compare cycle conditioning with explicit alignment, we explored this in the context of the diffusion model. Specifically, we attempted an alignment of the embeddings $E_{\theta}(S_0)$  and $E_{\theta}(S_t)$ directly using explicit loss (e.g. MSE between the representations of the clean and noisy samples). However, we observed that enforcing explicit equality between these embeddings leads to a collapse, where the network learns to produce a single embedding representation for all input samples. This is because the network can trivially satisfy the equality by mapping all inputs to the same output, which is undesirable and does not capture the variability that is necessary for robust representation learning. In contrast, our cycle conditioning approach avoids this collapse by supervising the latent representation with meaningful information from corresponding input samples.
> > >
> > > Include a brief preliminary explanation of time series with categorical data ...
> > >
> > > We have updated the paper to include this preliminary explanation. The explanation can be found in Page 1, first Paragraph.
> > >
> > >
> > > Provide relevant citations ... Without this, the method may appear overly ad-hoc ...
> > >
> > > We agree that combining several sub-modules (in our case, three) for a single task, such as extracting a meaningful and compact representation from an input sample, may seem ad hoc at first glance, as it requires specific design choices to ensure alignment between them. However, developing a complex system like our diffusion model that effectively utilizes feedback from the generative process to condition on the input sample is a challenging task. This is especially true because the model must both be dense and maintain consistency across different training iterations. Without such consistency, the diffusion model would ignore its conditioning information and, much like in the vanilla diffusion model, attempt to reconstruct the input based solely on the noise.
> > >
> > > To mitigate this, we introduced two additional modules to prevent the loss of conditioned information due to cycle conditioning, which helps guide the model towards more consistent outputs.
> > >
> > > Regarding the references, there are works that explore similar concepts for other tasks. For instance, methods related to latent motion learning (https://arxiv.org/abs/2203.09043, https://arxiv.org/abs/2305.06437, and https://arxiv.org/abs/2305.03989) use dictionary learning approaches to model human motions in videos. However, these methods focus on motion synthesis rather than the task of maintaining consistent conditioning across intermediate layers in a generative model. Similarly, works such as https://arxiv.org/abs/1607.04606 and https://arxiv.org/abs/1801.00632 explore character-level embeddings for text analysis. While our spelling supervision technique shares some superficial similarity to character-level feature extraction, the methods cited focus on extracting features at the character level and do not account for the order of letters to extract fixed-size vectors from time series with varying numbers of tokens at each time step.
> > >
> > > We believe our approach differs conceptually from the works mentioned and, including these references may confuse the readers rather than help contextualize our novel contribution. Thus, while there is some surface-level similarity in the ideas, we chose not to include these references as they are not directly aligned with the specific goals of our method.

---

> > > > ### Comment · Reviewer_FezA · 2025-01-28
> > > >
> > > > Thank you for the comprehensive response and the great effort. I now understand that each module is critical in preventing model collapse and enhancing robustness. The effectiveness of the proposed method, along with the extended ablation studies, compensates for the seemingly ad hoc nature of the design.
> > > >
> > > > However, regarding the authors' claim that "intermediate features in U-Nets are not inherently compact nor optimized to serve as standalone, low-dimensional representations without further processing," I believe these features can be easily adapted. For example, applying spatial pooling to transform a $16 \times 16 \times 256$ feature map into a compact 256-dimensional representation makes them highly suitable for downstream tasks, such as classification. (see https://arxiv.org/abs/2401.14404, https://arxiv.org/abs/2303.09769)
> > > >
> > > > Overall, I think the paper has merit and makes a valuable contribution to the field.

---

### Review · Reviewer_KxR8 · 2024-12-29

**Summary Of Contributions:**

This paper presents an innovative approach to categorical data representation using a diffusion-based method. The proposed solution integrates three core components: diffusion-based representation learning, cycle conditioning, and spelling supervision. These components work systematically to capture meaningful and general-purpose representations from categorical data, addressing key limitations of existing methods. The authors focus on healthcare datasets (heart failure and lung cancer prediction tasks), and demonstrate that their method outperforms state-of-the-art techniques, both in terms of AUROC and AUCPR.

**Audience:**

Yes

**Broader Impact Concerns:**

No concerns.

**Claims And Evidence:**

Yes

**Requested Changes:**

- Explicitly present and discuss ablation studies for cycle conditioning and spelling supervision in the main text to underscore their individual contributions to the overall performance, as indicated by the experimental results in Tables 6-7.
- Provide a detailed computational cost analysis, comparing the proposed method with other state-of-the-art techniques.
- Consider adding a table to summarize and compare the performance of different diffusion-based strategies from Figures 3–11.
- Address minor presentation issues:
  - The term "dense" in Figure 1 appears incomplete.
  - The formatting of tables is inconsistent; some lack top rules, while others lack bottom rules.
  - Figure 2 is unclear and seems to lack proper reference in the text.

**Strengths And Weaknesses:**

Summary of strengths

- The paper is well-structured, providing a detailed background on related work.
- The integration of cycle conditioning and spelling supervision into a diffusion-based framework is novel and addresses challenges in representation learning for categorical data.
- The proposed model outperforms existing state-of-the-art methods and other diffusion-based representation learning techniques.
- The paper introduces a novel data transformation strategy to capture meaningful information from sequences of categorical tokens, demonstrating superior performance compared to the BERT embedding layer.

Summary of weaknesses

- The evaluation focuses exclusively on the MIMIC datasets, which limits the applicability of the method to other domains like finance or recommendation systems.

- While the paper critiques the high computational demands of graph neural networks for categorical data, it does not provide a computational cost analysis of the proposed diffusion-based method. Given that diffusion models are computationally intensive, a detailed comparison of efficiency and scalability would be valuable.

---

> ### Author Response · Authors · 2025-01-01
>
> We thank the reviewer for the useful feedback. We address your comments below. We have carefully considered your suggestions and made the necessary revisions to improve the clarity and technical depth of the paper.
>
> Explicitly present and discuss ablation studies for cycle conditioning and spelling supervision...
>
> We have added a new subsection under Section 5 that explicitly explains the individual effectiveness of each submodule in our method (i.e., spelling supervision and cycle conditioning). We framed the discussion around two specific questions, addressing how each submodule contributes to the overall performance of the model.
>
>
> Provide a detailed computational cost analysis...
>
> We have included a new Section H, which explicitly compares the computational complexity, advantages, and disadvantages of our method with those of its competitors. This section provides a detailed analysis of the computational complexity of the two main parts of our method: the data transformation and the representation learning components. We believe this analysis highlights the trade-offs of our approach compared to other methods in terms of computational cost and efficiency.
>
>
> Consider adding a table to summarize and compare the ....
>
> A comparison between the performance of our method and its diffusion-based competitors is now presented in Table 6, where we provide a detailed comparison under different settings.
>
>
> Address minor presentation issues...
>
>  We also conducted a thorough revision of the figures and tables to ensure consistency and to address any cropping issues.

---

### Review · Reviewer_pmoU · 2025-01-05

**Summary Of Contributions:**

The paper proposes a new method to learn categorical data representations using diffusion models, aiming to learn more versatile and general-purpose information from categorical data and thus enhancing adaptability to unforeseen tasks. First, the paper proposes a simple data transformation strategy that maps categorical data at each time step to continuous data: a predefined vocabulary is used to construct a multi-hot representation for categorical data at each time step, and then the multi-hot representation is mapped to a fixed size representation through a learnable transformation matrix; the transformation matrix is optimized iteratively. Next, the paper proposes to further learn a compact representation of the transformed continuous data using diffusion models. Specifically, the paper learns a diffusion model conditioned on the compact representation, together with the encoder producing the compact representation. To encourage the extraction of general-purpose representations, the paper further proposes cycle conditioning which intuitively forces the encoder to capture the same content from both the clean and the noisy samples. To enhance the conditioning information, the paper proposes spelling supervision which ensures the latent condition can recover the "spelling vector" for each time series. Further, to encourage a denser latent code, latent space navigation is proposed to improve the supervision with spelling vectors. The paper then conducts representation learning on MIMIC III and MIMIC IV datasets and then uses the learned representations (mainly the latent code) to do disease prediction. The learned representations show SOTA classification performance. The paper also demonstrates probabilistic representation learning further enhances the representation, with the help of the stochastic nature of diffusion models. Lastly, ablation studies are conducted to show the effectiveness of cycle conditioning and spelling supervision.

**Audience:**

Yes

**Claims And Evidence:**

Yes

**Requested Changes:**

Minor ones:
1. The authors may consider correcting the citation formats, many citations should be in parentheses.
2. The authors may consider adding a definition of categorical data.
3. Section 4.1: in languages, typically sentences are also not of the same length and padding can be used to produce consistent fixed-size vectors for each sentence, so the argument that the embedding layer has "limitations when dealing with categorical time series" is not sound. Also, the argument that, training both the embedding layer and a continuous diffusion mode may not be stable, is not convincing - is the argument a conclusion from experiments?
4. In the paragraph above equation 10, it should be $M_\theta\left(g_\theta\left(S_t, t, E_\theta\left(\frac{1}{\sqrt{\beta_t}} S_t-\frac{\sqrt{1-\beta_t}}{\sqrt{\beta_t}} \epsilon_{t-1}\right)\right)\right)$ instead of $M_\theta\left(E_\theta\left(S_t, t, E_\theta\left(\frac{1}{\sqrt{\beta_t}} S_t-\frac{\sqrt{1-\beta_t}}{\sqrt{\beta_t}} \epsilon_{t-1}\right)\right)\right)$.
5. Figure 2 caption: it should be "left panel" in the last sentence.
6. Same concern in equation 12 as point 4 above.
7. Same concern in section 4.2.5 as point 4 above. Moreover, $g_\theta\left(S_t, t, E_\theta\left(\frac{1}{\sqrt{\beta_t}} S_t-\frac{\sqrt{1-\beta_t}}{\sqrt{\beta_t}} \epsilon_{t-1}\right)\right)$ is predicted noise, maybe a justification on why the predicted noise is useful to derive representations.

Critical ones:
1. Cycle conditioning: whether it is incorporated at all denoising time steps during training? Timesteps near pure noise may not be suitable since the added noise is too much and makes a big difference between clean and noisy data. In this case, incorporating cycle conditioning instead harm the learning of $E_{\theta}$. Hope the author could provide explanations on choices of time steps and ablation studies.
2. About the loss function in equation 14: each loss term should have a different magnitude and of a different importance level, I wonder why the authors decided to directly sum all of the loss terms without weighting, and it may be better to have a justification for this.
3. About the claim of learning "general-purpose information" using the proposed method: The downstream task is disease prediction (binary classification), how can this experiment results show learning of "general-purpose information"? It would be more convincing if the learned representations could support different types of downstream tasks.

**Strengths And Weaknesses:**

Strengths:
1. The method is novel for learning better representations for categorical data using the new powerful generation models - diffusion models.
2. The paper proposes careful and insightful design components to make sure meaningful representations are learned.
3. The proposed method shows superior performance.
4. The exploration of probabilistic representation learning is interesting, further highlighting the strengths of the learning method with diffusion models.

Weaknesses:
1. The paper has some typos, such as citation format and equations. See the next section.
2. The paper lacks some reasonable justification for certain design choices. See the next section.
3. The proposed method is effective on medical data and disease prediction, yet the effectiveness is unknown for other categorical data or downstream tasks.

---

> ### Author Response · Authors · 2025-01-10
>
> We thank the reviewer for the valuable feedback. We have addressed the comments as follows:
>
> The authors may consider correcting the citation formats....
>
> The authors may consider adding a definition of categorical data....
>
> We have updated the manuscript to correct the citation formats, ensuring that citations are placed in parentheses as required. Additionally, we have added a definition of categorical data, where we explain that in categorical data, each observation in a time series corresponds to a specific category rather than a numerical value. This lack of numerical continuity makes interpolation between observations not meaningful. We provide this definition and then proceed to explain how healthcare trajectories are categorical time series and why powerful representation learning is essential for such data.
>
>
> ...., so the argument that the embedding layer has "limitations when dealing with categorical time series" is not sound....
>
> We agree that large language models (LLMs) are powerful tools and that they handle varying sentence lengths effectively using simple padding strategies. However, there are two critical differences between neural language data (text) and categorical medical time series that make this approach less effective.
> First, in text, sentences are sequences of words where the order of words is always meaningful. If we treat text as a time series, there is only one token at each time step (a single word). In contrast, a categorical medical time series often includes multiple tokens at each time step, and the number of tokens can vary across time steps. Moreover, the order of tokens within each time step is typically not meaningful, as their arrangement is often arbitrary.
> Second, if we attempt to flatten a medical time series by concatenating all observations across time steps into a single sequence, similar to text, and then apply padding to manage the differing number of tokens, we risk introducing the randomness of token ordering within time steps into the inherently meaningful temporal ordering between time steps. For example, in a medical time series, the information at time step 1 is guaranteed to follow time step 0 in chronological order, but the tokens within each time step do not have such intrinsic ordering. This misalignment leads to critical limitations when using embedding layers from LLMs for categorical medical time series.
>
>
> Also, the argument that, training both the embedding layer and a continuous diffusion mode may not be stable, is not convincing.....
>
> Jointly learning the embeddings and the diffusion model itself has been a critical topic in recent years (https://arxiv.org/pdf/2211.15089, https://arxiv.org/abs/2212.11685), with several attempts to address it in the literature. These are two interdependent tasks: the optimal solution for the diffusion model requires an optimal embedding space as its starting point. However, when both tasks are learned simultaneously, several stability issues arise.
> Specifically, consider the process of projecting a time series into a continuous embedding space. Initially, this embedding is suboptimal and evolves over subsequent training iterations. When a training iteration of the diffusion model is introduced to learn a trajectory (e.g., a reverse path of 1,000 steps from noise back to this suboptimal embedding), the diffusion model essentially "learns" based on an embedding that is itself changing in subsequent iterations. This dynamic interaction dramatically enlarges the search space for optimal solutions, as both the embeddings and the diffusion model are trying to converge simultaneously.
> In the case of complex data, this exponentially expanded search space exacerbates convergence challenges. It may lead to significant instability in training or require a substantial increase in computational resources and time for the overall network to converge. These issues are not merely theoretical or even for embedding layers followed by diffusion models but have been observed in both in experimental setups and verified through analysis in different machine learning tasks like learning a warping map follwed by generating realistic images of the warped images and therefore have motivated methods in the literature to either use pre-train embeddings or decouple their optimization from the diffusion process.
>
>
> Regarding the typos....
>
> Thank you for your point. We have now corrected the equations.

---

> > ### Author Response · Authors · 2025-01-10
> >
> > Cycle conditioning: whether it is incorporated at all denoising time steps during training? ....
> >
> > Thank you for raising an important point regarding this issue.  In response to your query, cycle conditioning is indeed applied across all timesteps during training. Based on our experiments, we have found that this approach does not harm the learning process, even at higher noise levels.  We would like to clarify why this is the case. One important aspect of our method is that it does not directly minimize the distance between the embedding of a clean condition and the embedding of its corresponding clean sample Instead, we aim to leverage the generative power of a diffusion model to extract features from the conditioning sample while actively evaluating whether the conditioning is effective, based on feedback from the generative process itself.
> > To elaborate further, the primary loss function in our approach is two MSE loss functions between the predicted noise and the true noise, when our model is conditioned using two versions of the input sample. No loss function is directly applied to the embedding space itself (though cross-entropy losses are applied using a latent navigation strategy).
> > We also fully acknowledge that, even for the MSE loss of diffusion models, the noise levels at later timesteps can be challenging to handle. In such cases, the extracted noise from a sample with too much noise could potentially hinder the learning process by preventing it from outputting a correct and unique solution. Indeed, this challenge is well-known in the literature, and several strategies, such as weighted learning, have been proposed to address this issue.
> > However, as the forward process in our model is based on introducing Gaussian noise, which ensures that the transformed samples are ultimately Gaussian, this prior knowledge helps make the task tractable and simplifies learning, even at higher noise levels. Thus, despite the noise, the approach remains effective.

---

> > > ### Author Response · Authors · 2025-01-10
> > >
> > > About the loss function in equation 14 ....
> > >
> > > Thank you for your insightful comment. We recognize that our notation involving summation over the means of the samples might have been misleading, as it suggested that we explicitly minimize the difference between the posterior and prior means. However, in practice, the minimization is based on the difference between the true and estimated noise added to the samples, which is mathematically equivalent to minimizing the distance between the posterior and prior distributions of the samples (https://arxiv.org/abs/2006.11239).
> > > To address this concern, we have removed the potentially confusing notation and clarified the process in the main text, specifically under the section "Learning," where we explain the multi-step optimization strategy we employed. This approach ensures consistency in the magnitude of the loss terms. It is important to note that the magnitude of the loss function for estimating the noise remains the same across the two conditions, which is why we sum the two loss terms without any additional weighting.

---

> > > > ### Author Response · Authors · 2025-01-10
> > > >
> > > > About the claim of learning "general-purpose information" ....
> > > >
> > > > Thank you for your insightful point. We agree that the term "general-purpose" can have a broad meaning and can be interpreted in different ways. However, in the context of medical disease prediction, especially tasks such as predicting whether a patient will develop a disease in the future, it is a significant challenge to extract features that are not overly tailored to one specific disease. For example, predicting both heart failure and lung cancer, two highly unrelated diseases, requires the model to learn representations that generalize well across such diverse tasks, rather than optimizing for one task at the expense of the other.
> > > > In light of your suggestion, we have removed the term "general-purpose" and replaced it with more precise wording "multi-task". Additionally, we agree that evaluating the performance of our method across different types of downstream tasks would strengthen our argument. We plan to carry out such evaluations. However, planning, and coding for these additional tasks will require time. We intend to explore these experiments in the near future.

---

### Author Response · Authors · 2025-01-10
**General Response**

We would like to thank all reviewers for their constructive feedback. We greatly appreciate the time and effort invested in reviewing our work, and we hope that our responses adequately address your concerns.

As highlighted by the reviewers, our work introduces a novel approach to leveraging diffusion models for representation learning from categorical data. Specifically, we propose a data transformation strategy that learns to extract fixed-size embeddings from each time step of a time series, even when the number of tokens (observations) per time step varies. Through this approach, we demonstrate the effectiveness of our method on medical time series data, such as the MIMIC database.

Additionally, we present a novel approach to transform the latent features of the conditioning sample in a diffusion model into a versatile, general-purpose representation. This design leverages the compact latent space of the conditioning sample, enabling diffusion models to function as effective representation learning networks. This contrasts with traditional methods, which attempt to modify intermediate layers of the U-Net architecture for dimensionality reduction. Such strategies are less efficient, as reducing the size of the intermediate layer compromises the U-Net's ability to reconstruct input samples. Conversely, maintaining a large intermediate layer results in a representation that is too extensive to be considered compact or general-purpose. Our method addresses these limitations in an effective way.\ To enhance the clarity and quality of our work, we have updated the manuscript (with changes highlighted in blue), including additional discussions and experiments as requested.

---

### Comment · Editors_In_Chief · 2025-03-02
**Camera Ready still anonymous**

Hi authors,

I just noticed that the submitted camera-ready version does not match the required specification -- most significantly, it should be non-anonymized. You were supposed to submit the non-anonymized version previously when solicited, and the AE was supposed to check to make sure it looks right before verifying, neither of which seem to have happened.

Can you please email tmlr-editors@jmlr.org with the proper, final, non-anonymized version of the paper?

Gautam

---

### Decision · Action_Editor_GEJr · 2025-02-15

**Recommendation:** Accept as is

**Comment:**

The paper introduces a novel diffusion-based approach for categorical data representation. However, its evaluation is limited to the MIMIC datasets, raising concerns about the method’s generalizability to other domains.

**Audience:**

The studied problem is of sufficient interest to the TMLR community.

**Claims And Evidence:**

This paper introduces a diffusion-based framework for representation learning in categorical (discrete) time series data. The proposed approach consists of two primary components: (1) a data transformation module that converts multi-hot token representations at each time step into continuous vectors while maintaining token set similarities, and (2) a diffusion-based representation learning module that reconstructs the original sequence and extracts robust features as conditioning embeddings. This framework achieves significant improvements over state-of-the-art methods, particularly in predictive healthcare applications using the MIMIC-III and MIMIC-IV datasets. Additionally, the authors conduct comprehensive ablation studies, demonstrating the critical role of each module in preventing model collapse and improving robustness, thereby addressing concerns about the seemingly ad hoc nature of the design.

The claims in the paper are mostly well-supported by experimental results.